# Confinement of air in the Asian monsoon anticyclone and pathways of convective air to the stratosphere during summer season

Bernard Legras and Silvia Bucci

Laboratoire de Météorologie Dynamique, UMR CNRS 8539, IPSL, PSL-ENS/Sorbonne Université/Ecole Polytechnique, Paris, France

**Correspondence:** Bernard Legras (bernard.legras@lmd.ens.fr)

**Abstract.** We study the transport pathways from the top of convective clouds to the lower tropical stratosphere during the Asian monsoon, using a dense cover of Lagrangian trajectories driven by observed clouds and the two reanalyses ERA-Interim and ERA5 with diabatic and kinematic vertical motions. We find that the upward propagation of convective impact is very similar for the kinematic and diabatic trajectories using ERA5 while the two cases strongly differ for ERA-Interim. The parcels that stay confined within the Asian monsoon anticyclone and reach 380 K are mostly of continental origin while maritime sources are dominating when the whole global 380 K surface is considered. Over the continent, the separation of descending and ascending motion occurs at a crossover level near 364 K which is slightly above the clear sky zero level of radiative heating rate, except over the Tibetan Plateau. The strong impact of the Tibetan Plateau with respect to its share of high clouds is entirely due to its elevated proportion of high clouds above the crossover. The vertical conduit found in previous studies actually ends where the convective clouds detrain. Subsequent parcel motion is characterized by an ascending spiral that spans the whole anticyclone. The mean age of parcels with respect to convection exhibits a minimum at the centre of the Asian monsoon anticyclone, due to the permanent renewal by fresh convective air, and largest values on the periphery as air spirals out. This contrast is reduced by dilution for increasing altitude. Above 360 K, the confinement can be represented by a simple 1D process of diabatic advection with loss. The mean loss time is about 13 days and uniform over the range 360 K to 420 K which is to be compared with a total circulation time of two to three weeks around the anticyclone. The vertical dilution is consequently exponential with an e-folding potential temperature scale of 15 K (about 3 km). The mechanism is compatible with the appearance of a columnar tracer pattern within the anticyclone. It is noticeable that the tropopause does not exhibit any discontinuity in the transport properties when seen in terms of potential temperature.

## 1   Introduction

The Asian monsoon is the most active convective region during boreal summer and, as such, is also the largest provider of air ascending from the tropospheric boundary layer to the upper troposphere and the lower stratosphere. Usually the air does not ascend directly into the stratosphere but convective air is mostly detrained below the tropopause within the tropical tropopause layer (TTL) and is subsequently carried aloft by slow motion (Fueglistaler et al., 2009) . The bulk of the convective detrainment occurs at about 200 hPa or 350 K (12-13 km) and is associated with the divergent upper component of the

Hadley circulation (Garny and Randel, 2013). At such altitudes, radiative cooling dominates and the vertical motion is almost everywhere descending except within the clouds. It is only at about 355-360 K (14-15km), depending on the location, that the mean clear sky radiative cooling leaves room for warming (McFarlane et al., 2007). As the short-wave absorption is very small, the reversal is mainly a long-wave effect due to the very cold temperature at the tropopause, such that the absorption of upward long-wave radiation exceeds the emission. This cold temperature is maintained by the adiabatic cooling of the air which is pumped across the tropopause and enters the Brewer-Dobson circulation (Randel et al., 2007; Randel and Jensen, 2013). The radiative effect of clouds on their environment provides a perturbation that can, according to its sign, rise or lower the local mean level of zero radiative heating (LZRH) (Yang et al., 2010; Wright and Fueglistaler, 2013; Berry and Mace, 2014; Johansson et al., 2015). As the LZRH plays the role of a repelling transport barrier, a question is whether it forbids air parcels detrained below to reach levels above. Corti et al. (2006) proposed that fluctuations in the radiative heating provided by cirrus clouds may help parcels to cross the LZRH. However, Tissier and Legras (2016) found that this seldom happens and that convective sources of air reaching the tropopause are for 80% located above the all sky LZRH.

During the Asian monsoon, a wide anticyclonic circulation denoted as the Asian Monsoon Anticyclone (AMA) sits over the quasi-stationary low pressure centre in the lower layers. This circulation, which reaches its maximum intensity near 360 K, ventilates the top of the monsoon convective clouds and redistributes the detrained air over a large area, favouring latitudinal transport (Dunkerton, 1995; Dethof et al., 1999). Satellite observations show that tropospheric compounds emitted at the surface, like CO, and aerosols generated from these compounds, tend to concentrate within the AMA while it is depleted in stratospheric borne ozone (Randel and Park, 2006; Park et al., 2008; Randel et al., 2010; Vernier et al., 2015; Santee et al., 2017; Luo et al., 2018). These observations have been corroborated by a number of numerical simulations based on Lagrangian or general circulation models which all show confinement of tropospheric tracers within the AMA (James et al., 2008; Park et al., 2009; Tzella and Legras, 2011; Wright et al., 2011; Bergman et al., 2012, 2013; Orbe et al., 2015; Vogel et al., 2015; Yan and Bian, 2015; Tissier and Legras, 2016; Garny and Randel, 2016; Pan et al., 2016; Fan et al., 2017; Ploeger et al., 2017; Vogel et al., 2019; Wu et al., 2020). There is, however, a lack of consensus on the interpretation of what is actually observed. Basically, some studies support that the ascent of air within the AMA, in the clouds and above the clouds, canalizes the flow from the troposphere to the stratosphere in a sort of isolated chimney with a core above the Tibetan Plateau. This upward flow would be then redistributed horizontally at the top of the chimney, corresponding approximately to the tropopause level above the Tibetan Plateau (Bergman et al., 2013; Pan et al., 2016; Ploeger et al., 2017). Other studies see a broader spiralling ascent and stress that only a limited part of the ascending flux is processed within the AMA and that a large flux, mostly of maritime origin, finds its way to the stratosphere by circulating around the AMA without penetrating its core (Orbe et al., 2015; Tissier and Legras, 2016; Fan et al., 2017; Vogel et al., 2019). It has been discussed that some of these discrepancies can be due to the differences between the reanalysed winds, vertical velocities and heating rates which are quite different among available reanalyses, being notoriously large between ERA-Interim/IFS and MERRA-2/GEOS5, the most used by modellers, in the Asian monsoon region (Tegtmeier et al., 2019).

The goal of this work is to seize the opportunity of the release of the ERA5 by the European Centre of Medium Range Weather Forecast (ECMWF), a new generation reanalysis incorporating very recent progresses in numerical weather forecast,

to revisit this problem in a systematic and quantitative way. In addition, we use state of the art diagnostics of the heights of convective clouds from high frequency and high resolution geostationary observations. We also focus on reconciling previous studies and on providing a simple quantitative account of the modelled transport across the TTL with a 1D model of transport with losses.

Section 2 describes the data and methods used in this study. Section 3 presents the results of the extensive 3D Lagrangian
calculations. Section 4 discusses how a simple 1D model can reproduce the properties of the 3D transport across the TTL. Section 5 offers discussion and conclusion. Complementary results and discussions can be found in the Supplement.

## 2    Data and methods

### 2.1    ECMWF reanalyses

We compare here the two reanalyses ERA-Interim and ERA5 of the ECMWF. The ERA-Interim was initially made available
at the end of 2007. It is based on the Cy31r2 version of the Integrated Forecast System (IFS), released in 2006, with T255L spectral resolution and 60 hybrid levels up to 0.1 hPa (Dee et al., 2011). It uses a 4D-Var assimilation with a 12h cycle. The ERA5 reanalysis (Copernicus Climate Change Service, 2017; Hersbach et al., 2020) was initially made available at the end of 2017. It is based on the C41r2 version of the IFS model with T699L spectral resolution and 137 hybrid levels up to 0.01 hPa. It uses an ensemble 4D-Var assimilation with a 12h cycle. Between 155 and 65 hPa, the number of hybrid levels has
changed from 6 to 17, providing a much better vertical resolution of the TTL. The IFS has undergone many changes between ERA-Interim and ERA5, in particular regarding the parametrization of cloud processes.

We use ERA-Interim winds, temperature and radiative heating rates on the model hybrid levels at a $1°$ horizontal resolution in latitude and longitude over the sphere. Winds and temperature are available 3-hourly by interleaving the 3h and 9h forecasts with the 0h and 6h analyses during each 12h assimilation cycle. Radiative heating rates, produced by the forecast, are avail-
able 3-hourly. We use ERA5 winds, temperature and radiative heating rates on the model hybrid levels at a $0.25°$ horizontal resolution in latitude and longitude over the domain described in Sec. 2.3. The data are available hourly from analysis (winds and temperature) and forecasts (heating rates). Notice that in both ERA-Interim and ERA-5, the winds and temperature are instantaneous fields while the radiative heatings are accumulations over, respectively, 3h and 1h. The heating rates are treated as centred with a time shift with respect to winds and temperature, of 1h30 in the ERA-Interim and 0h30 in the ERA5. The
vertical shift of the vertical velocities with respect to the other variables in the IFS is also accounted.

The ERA-Interim and the ERA5 differ significantly in the cloud properties over the Asian monsoon region (Tegtmeier et al., 2019). The maximum cloud cover occurs at a lower level in the ERA5 than in the ERA-Interim and is smaller. The ERA5, however, exhibits more high penetrative convection than the ERA-Interim, especially over the Tibetan Plateau (Tegtmeier et al., 2019) and more activity at better resolved small scales in general (Hoffmann et al., 2019). Fig.1 shows the cloud radiative
heating (CRH), defined as the difference between the all sky and the clear sky radiative heating, for both reanalyses in the monsoon region. The cloud radiative heating is exerted at lower altitude in the ERA5 and therefore disturbs less the LZRH from its clear sky value than in the ERA-Interim. The ERA5 cloud disturbance of the LZRH concentrates over land and in

particular over the Tibetan Plateau which corresponds to the largest lobe in the green curve. Tegtmeier et al. (2019) found that the all sky ERA5 radiative heating rates are the most consistent with estimates based on active satellite measurements of the cloud vertical distribution (Berry and Mace, 2014; Johansson et al., 2015) when compared with four other reanalyses .

## 2.2 Lagrangian trajectories

Lagrangian trajectories are calculated using the Lagrangian model TRACZILLA (Pisso and Legras, 2008), which is a variation of FLEXPART (Stohl et al., 2005). TRACZILLA interpolates velocities and heating rates directly from the hybrid grid to the location of the parcel using vertical log pressure interpolation for kinematic trajectories or potential temperature interpolation for diabatic trajectories. The time step is 7.5 minutes. In both cases, temperature and pressure are calculated along the trajectory. As the trajectories are intended to wander at high altitude outside the clouds, the diabatic calculations use only the all sky radiative heating rates. The kinematic calculations use the total vertical velocities that includes the net convective flux, which cannot be separated. This set-up is similar to that of Bergman et al. (2013). Some other studies where the tracers are initialized in the boundary layer used instead the total heating rate (Vogel et al., 2015, 2019; Li et al., 2020). As the altitude rises above the level of maximum cloud cover, the total heating rate converges rapidly to the radiative heating rate. Experiments made using total diabatic heating (not shown) do not change our results.

## 2.3 Domain

The confinement properties of the AMA are easily established from the strong correlation of time-averaged maps of tropospheric tracers and the AMA mean circulation (Park et al., 2007, 2009; Randel et al., 2010). The mean Montgomery potential is shown in Fig. S2 of the Supplement on several isentropic levels and defines the extent of the AMA during summer 2017. However, the AMA fluctuates a lot from day to day and it is very difficult to define an operational boundary on instantaneous maps. Ploeger et al. (2015) use the gradient of potential vorticity to mark the boundary of the AMA but this boundary is very often fuzzy and broken into numerous patches. We avoid this problem by considering a domain, denoted as FullAMA, that is intended to encompass the AMA and reveals its confinement properties but, at the same time, to be small enough such that trajectories leaving the AMA also leave the domain very shortly after. The FullAMA domain is bounded in longitude by $10°$W and $160°$E and in latitude by $0°$N and $50°$N. This choice is also dictated by practical considerations as it is very costly to manage calculations using the ERA5 at full resolution in the global domain.

Therefore, the ERA5 calculations are conducted within the FullAMA domain, all trajectories reaching its boundary being discarded. For the sake of comparison, the ERA-Interim calculations are conducted both in the global domain and the FullAMA domain, the latter configuration being obtained by clipping the global trajectories which leave FullAMA. In addition, all trajectories reaching the 30 hPa or 500 hPa surfaces are discarded since parcels then exit the vertical region of interest. The global trajectories of the ERA-Interim are also exploited to study the impact of the monsoon at planetary scale.

## 2.4 Cloud data and trajectory sources

We characterize cloud tops using the SAF-NWC (Eumetsat Satellite Application Facility for Nowcasting) software package (Derrien and Le Gléau, 2010; Sèze et al., 2015) that determines cloud type and cloud top height from geostationary satellites using visible and infra-red channels. The 2016 version of the SAF-NWC package has been applied to the MSG1 (Meteosat 8) and Himawari satellites with 15 minutes sampling for MSG1 and 20 minutes sampling for Himawari during June-July-August 2017. MSG1 is used west of 90°E and Himawari east of this longitude. The auxiliary temperature profiles are provided from the ERA5 at hourly temporal resolution and 32 pressure levels, as well as the altitude and the temperature of the thermal tropopause. The cloud data at full spatial resolution are projected onto a regular grid in the FullAMA domain with spatial resolution 0.1° in both longitude and latitude using the closest neighbour method. The data are updated on this grid every 5 minutes to the most recent satellite data. When a satellite image is missing, the gap is filled by the last image available for this satellite. The SAF-NWC package uses an ensemble of retrieval algorithms choosing the best one for each pixel according to a selection tree. See Bucci et al. (2019) for a more detailed account of the method. Depending on the retrieval path, the cloud top pressure can be determined among a continuous range or within a set of discrete values. In particular, a small number of single pixel cloud tops are found at 100 and 70 hPa. We have not filtered these values as they are found at the core of very high and cold systems and are liable to capture overshooting events. No convective clouds are considered outside of the FullAMA domain.

In forward runs, cloud tops are used to initialize trajectories every hour over the period June-July-August 2017 by selecting high clouds in the FullAMA domain. We retain all cloud pixels above 250 hPa within high, very high opaque and thick semi-transparent types as per the SAF-NWC classification. For each of the selected cloud pixels, a new trajectory is launched at its top and is integrated forward in time for two months. Notice that a large number of mid-level clouds which are associated with heavy monsoon precipitations escape this selection. Four separate integrations of two months are performed. The first two uses the ERA5 dataset and are bounded within the FullAMA domain. In the sequel, the diabatic version is labelled EAD and the kinematic version is labelled EAZ. The additional two integrations are performed using the ERA-Interim dataset and the trajectories are integrated within the global domain. The diabatic version is labelled EID and the kinematic version is labelled EIZ. On the overall, 1.2 billions trajectories are launched during summer 2017.

In backward runs, the trajectories are initiated on a one degree grid at selected potential temperature and are launched every 15 minutes in the FullAMA domain, and every hour in the global domain for July-August-September 2017. The trajectories are integrated backward for up to two months when they do not exit the domain. The trajectories are then processed to find encounters with clouds. This is done, for each parcel, from 6-hourly outputs by interpolating the parcel position every 5 minutes and comparing the parcel pressure with that of the cloud tops from the SAF-NWC image at 0.1° resolution which is valid at that time. When a cloud top with lower pressure than the parcel is found at the same location, the backward trajectory is flagged as ending there.

## 2.5 Convective impact

The effect of the transport of monsoon convective parcels in forward calculations is measured as the convective impact. Basically, we divide the tropical atmosphere into cells of width $D = 1°$ in latitude-longitude and of height $\Delta\theta = 5$ K. We then count the number of convective parcels found within each cell over the full two months of integration. This count can be performed in the *target space*, that is at the location of the parcels when they are sampled along their trajectories, or in the *source space*, that is at the location of the parcels when they are released. In the latter case, the parcels are sorted according to their potential temperature in the target space. In order to be independent of the arbitrary discretization, the convective impact is weighted for each convective parcel by $\tau\delta^2\cos(\phi_S)$ where $\tau$ is the time interval between two satellite images (1 hour), $\delta$ is the size of the pixel in the satellite image (0.1°) and $\phi_S$ is the latitude of the convective source. In the target space, the count is further multiplied by the sampling interval along the trajectory $\Delta t = 6$ hours and is normalized by the mesh size in the target space that is $\Delta\theta D^2\cos(\phi_T)$ where $\phi_T$ is the mean latitude of the target cell. Hence the full normalization factor for the impact in the target domain is $\tau\Delta t\delta^2\cos(\phi_S)/\Delta\theta D^2\cos(\phi_T)$. The resulting quantity is called the *impact density*. In the source domain, $\phi_T$ is replaced by $\phi_S$ in the denominator and therefore the cosine factors disappear. The resulting quantity is called the *source density*. We define the *cumulated impact* as the integral of the impact density over the FullAMA domain for a given level. A convenient unit for the impact density is $\mathrm{day}^2\mathrm{K}^{-1}$ and we use $\mathrm{day}^2\mathrm{km}^2\mathrm{K}^{-1}$ for the cumulated impact

As the impact density and the source density vary considerably with altitude, it is useful to define an equalized quantity for the sake of comparison. The *equalized impact* is defined , for each cell within a given domain, as the ratio of the impact density to the value obtained by redistributing equally the cumulative impact over all cells, according to their area. We stress that each forward parcel is allowed to be counted as many times as it appears within the domain in the 6-hourly outputs during the integration period.

The backward trajectories are analysed according to the probability of hitting a convective cloud within the integration time or exiting the FullAMA domain. The sources are counted on a mesh of 1° resolution in the horizontal and 1 K in the vertical. They are normalized and equalized in the same way as the forward sources. An important difference with respect to the forward case is that in the backward analysis, only the first hit is accounted.

The impact is not meant to be an estimate of the convective mass flux or the mixing ratio of convective air since we do not have, in the present state of the art, any information on the mass detrainment at the top of the observed clouds. It should be seen at a metric for convective influence which can be used to study how parcels originated from convection are confined, dispersed or diluted in the TTL.

## 3 Results

### 3.1 Impact overview

We first present an overview of the impact in order to justify our approach. We use the ERA-Interim diabatic trajectories (EID) to compare calculations made in the global domain and in the restricted FullAMA domain. Figure 2 shows the impact density on

the 380 K surface for parcels launched from the cloud tops over the period 1st June - 31 August 2017 and integrated forward for two months. The confinement of air masses inside the AMA is clearly visible and exhibits similar patterns in both the FullAMA and the global domain versions of EID (panels a & c). Over the FullAMA domain, Figure 3 shows that the correlation of the impact at 380 K between these two versions is 94.3% and on the average is 96.9% between 340 K and 420 K (black solid curve). The ratio between the maximum impact in Figure 2(a) and Figure 2(c), is 0.91. This ratio decays from 1 at 350 K to 0.8 at 420 K (red solid curve in Fig. 3). Similarly, the ratio of the cumulated impacts over the FullAMA domain is 0.6 at 380 K and decays from 0.88 at 350 K to 0.52 at 420 K (blue solid curve in Fig. 3). At the same time, the cumulated impact itself (shown in Fig. 4) decays by a factor 18 in the global calculation and 30 in the FullAMA calculation. The differences between the two calculations are due to parcels that leave the domain and disappear in the FullAMA calculation while they re-enter the FullAMA domain in the global calculation. However, these parcels seldom re-enter the AMA core, hence the difference between the ratios of the maxima and of the cumulated impacts. The reentering parcels have travelled over long loops or even a full latitude circle and have been carried within filamentary structures submitted to strain and stretching resulting in mixing with the environment. Therefore, it is arguable to count them within the background rather than being part of the convective parcels when they reenter the FullAMA domain. We conclude that the confinement seen in the FullAMA calculations is not an effect of the boundaries and we will focus on this domain in most of the sequel. Further comparisons between the FullAMA and the global domain are made in Sec. S7 of the Supplement .

Figure 2 also shows the FullAMA impact for the ERA5 diabatic trajectories (panel e). The pattern is very similar to that of the ERA-Interim trajectories in the panel (a) with a correlation of 99%, but the maximum impact ratio and the cumulated impact ratio are, respectively, 0.44 and 0.46. These results are obtained because the horizontal distribution of the impact depends essentially of the horizontal isentropic circulation which seldom differs between ERA-Interim and ERA5 while the amplitude ratios depend on the vertical motion which differs a lot over the whole column as shown in Fig. 3 (dash curves).

Finally, Figure 2 shows the source density of convective parcels reaching the 380 K surface (panels b,d,f). The largest contribution in the FullAMA ERA-Interim trajectories (panel b) arises from North India (mostly the Ganges valley, Bengal and the north of the Bay of Bengal), with two other spots over the south of the Tibetan Plateau and in South China. These areas are surrounded by a wide halo of sources mainly over the Asian continent but with some significant component over the Sea of China and the Pacific east of the Philippines. The distribution of the global sources (panel d) exhibits a general intensification, by about a factor 2, without changing the pattern of continental sources but also a shift towards a larger share of the maritime sources, that are much more intense relative to the continental sources. The trajectories from maritime sources extend along the easterlies in the southern branch of the AMA and mostly leave the FullAMA domain without recirculating around or entering the AMA core.

The source distribution for the ERA5 is similar to that of the ERA-Interim but with important differences. The distribution is more concentrated on the Ganges valley and the Tibetan Plateau with weakened contributions from Bengal and South China. The source distribution shows a relative minimum over the narrow region which corresponds to the steep southern slope of the Himalayan . The monsoon flow hitting this slope generates a lot of precipitations but does not lead to high convective towers

that penetrate the TTL. More generally, other areas providing a lot of monsoon precipitations like the Ghats in South India or the Arakan mountains are not visible in our source maps.

## 3.2 Vertical transport and erosion

The upper panel of Fig. 4 shows the cumulated impact for the four FullAMA experiments (EAD, EAZ, EID, EIZ) compared to the high cloud distribution which is the common initialization for all experiments. The cloud distribution strongly peaks at $\theta = 349.5$ K and is mostly distributed between 340 K and 370 K. Some rare convective events, however, are still found up to about 400 K in the stratosphere while the applied 250 hPa selection threshold produces a cut-off in the lower layers below 335 K. The cumulated convective impact in the FullAMA domain peaks near the cloud peak. The peak is located lower by a few degrees for the diabatic trajectories (solid) than for the kinematic trajectories (dash) and the total impact below the cloud peak is also larger. The EID impact curve (solid red) exhibits the smallest maximum and is also associated with the smallest slope above the source peak, indicating that this case corresponds to the fastest upward transport. On the opposite, the EIZ impact curve (dash red) exhibits a large maximum and the largest slope. The two ERA5 cases (blue) exhibit intermediate and similar slopes. We already see here, as it is confirmed below, that EID and EIZ calculations exhibit large differences and bracket the two ERA5 calculations which are much closer. The lower panel of Fig. 4 shows the distribution of the clouds together with the vertical profile of the heating rate, here reduced to the restricted domain (20°E-140°E and 10°N-40°N) which avoids the frontiers of the FullAMA domain. We see that the zero level of radiative heating is above the maximum level of the sources, in a range of altitudes where the cloud density basically decays exponentially with potential temperature. The straight line in the panel corresponds to a decrement rate $\beta = 0.325$ K$^{-1}$. As the ERA-Interim all sky heating crosses the zero axis at a lower altitude than ERA5 and provides stronger heating above 370 K, the ERA-Interim impact is expected to be stronger and to propagate faster upwards, as observed.

We now define the age of a parcel as the duration elapsed from its release by convection. Figure 5 shows how the convective impact propagates inside the FullAMA domain from the sources as a function of age. In both the kinematic and diabatic cases, the dispersion occurs upward and downward. A clear separation occurs in the diabatic cases between the ascending and the descending branches, transporting parcels away from the main source level. The descending diabatic branches are very intense, ending at the imposed cut-off level. The upper branches exhibit a strong attenuation due to parcels exiting through the lateral boundaries, leading to the strong decay of the impact with altitude seen in Fig 4. The upward propagation is the slowest for EIZ and the fastest for EID with the two ERA5 cases in between and fairly close together. The propagation is estimated by fitting a straight line to the crests of the distribution on each isentropic layer. See also Sec. S4 of the Supplement. The slope, denoted as $A$ in the sequel, is found to be 1.08, 1.11, 0.97 and 1.35 K day$^{-1}$ for, respectively, EAZ, EAD, EIZ and EID. For diabatic cases, these values are consistent with the average heating rates of ERA5 and ERA-Interim above 370 K (see Fig. 4). It is shown in Sec. 4 that this result is expected when diffusive transport by heating rate fluctuations can be neglected. The descending branch of the diabatic impact and the withdrawal of parcels from around the 350 K level is also consistent with the heating rate crossing from cooling to warming near this level in both reanalyses. The close proximity of the zero level of

heating rate and of the maximum detrainment of the clouds is not fortuitous but can be seen as a manifestation of the Fixed Anvil Temperature principle (Hartmann and Larson, 2002).

Even if the vertical velocities are ascending everywhere in the monsoon region, the kinematic distributions in the two left panels of Fig. 5 exhibit a clear descending branch. The descent does not occur within the monsoon region but as a result of the westward horizontal transport by the AMA that brings the convective parcels over the Arabian desert and the Sahara where the vertical motion is subsiding (see Sec. S5 of the Supplement). This transport is one of the branches of the Hadley-Walker circulation during the monsoon season. As we see in the next section, it is also produced by the diabatic circulation. The large differences between kinematic and diabatic trajectories transport properties in the ERA-Interim and the fast vertical transport of its diabatic version have already been noticed in a number of previous studies (e.g., Ploeger et al., 2012; Bergman et al., 2015; Bucci et al., 2019; Li et al., 2020).

Figure 6 shows the decay of the total impact integrated over the FullAMA domain and three vertical layers (defined in the caption) as a function of age. At small ages, the impact is confined in the mid-layer, where the sources concentrate. Then, the upper and lower layer impacts grow, and the latter is rapidly dissipated by the bottom cut-off, so that eventually the upper layer impact dominates. The decay of the lower and mid layer impacts is much faster in the diabatic cases due to the descending motion within the source region. The asymptotic decay time scale $\alpha$ of the upper layer is quite similar among EAD, EAZ and EID. It is larger for EIZ but the asymptotic limit, where the upper layer dominates, is only marginally reached in that case. We retain the value for ERA5 diabatic, that is $\alpha = 13.3\,\mathrm{day}$ as the erosion time-scale of the upper-layer. This time scale is of the order or smaller than the mean circulation rate in the AMA, two to three weeks, as found in Sec. S2 of the Supplement. Therefore the AMA exhibits only weak confinement properties. If we assume that this erosion rate and the mean vertical ascent $A$ explain the dilution of the impact with altitude, we get a decay rate $(\alpha A)^{-1} = 0.068\,\mathrm{K}^{-1}$ very close to the value $0.065\,\mathrm{K}^{-1}$ obtained from Fig. 4.

In the sequel, we will focus on the ERA5 diabatic trajectories This set-up is shown to be the most relevant to interpret the airborne data of the StratoClim campaign by Bucci et al. (2019) and it produces results very close to the kinematic set-up as far as the upper branch of transport from the convective sources is concerned. More comparisons between diabatic and kinematic trajectories and between FullAMA and global trajectories can be found in the Supplement.

### 3.3 Horizontal distribution of impact and sources

In order to investigate how the confinement varies with altitude, Fig. 7 shows the impact distribution for four layers from 340 K to 370 K in the ERA5 diabatic calculations. In the lowest layers, at 340 K and 350 K (panels a and c), the convective parcels are rapidly expelled to other regions by the divergent motion which is maximum at these levels (see Fig. S4 of the Supplement) and to lower levels by diabatic cooling. Due to this combined effect, the impact at 340 K is maximum west of the monsoon region over the Arabian desert and the Sahara where the air is slowly subsiding. This upper circulation from the monsoon uplift region is one of the main branches of the Hadley-Walker circulation during summer. Other descent regions in the Southern Hemisphere, corresponding to the other branches, appear for global trajectories (see Fig. S10 of the Supplement). We stress that the descent branches are also observed for kinematic trajectories, as seen in Figs. S8 and S10. However, as

vertical velocities are positive over the whole tropospheric column in the monsoon region, the total descending impact is reduced with respect to the diabatic simulations. The distribution of sources for the 340 K and 350 K levels (see Fig. 7(b,d)) has a large maritime contribution, in particular over the Bay of Bengal (BoB), and the continental contribution concentrates over India in the eastern regions adjacent to the BoB.

A drastic change in the pattern occurs at 360 K with the impact now centred over continental Asia (see Fig. 7(e)) and a distribution of the sources concentrated over North India and the Tibetan Plateau with maritime sources only on the North of the BoB (see Fig. 7(f)). Figure 7(g,h) and Fig. 8 show that between 370 K and 420 K, the pattern remains basically constant both for the target impact and for the sources, but for the exponential dilution shown in the upper panel of Fig.4, which is fully explained above as the combination of constant loss with uniform ascent over the AMA. It is remarkable that the impact pattern, in spite of the dilation due to pressure drop (that cannot occur in the vertical because of the quasi-constant ascent rate), does not exhibit any expansion with height and rather shows a columnar shape. The target impact contours closely follow the contours of the Montgomery potential (see also Fig S2) that describes the main circulation within the domain.

The self similar behaviour of the forward impact is confirmed by the backward trajectories. Fig. 9 is produced from the backward EAD trajectories for the three months July-August-September 2017 within the FullAMA domain. The percentage of convective hits reaches 100% at 360K within a large region at the centre of panel (a). This percentage decreases with altitude but remain high, with values above 80%, up to 380 K, and up to 50% in the AMA core at 400 K while the overall shape remains the same at all levels. The region of high percentage of convective hits is surrounded by a periphery which is almost entirely filled with background air that came across the boundary of the FullAMA domain. Figure 9 also shows the source distribution in the horizontal and in the vertical. The parcels released at 360 K reach convection in a close neighbour of this level (panel c) and they show a mixed distribution of sources over the maritime and continental regions (panel b) like in Fig 7(f). For parcels released at higher levels the vertical distribution of sources broadens from near 360 K to the level of release (panels f, i, l). The distribution tends to peak near 365 K and is skewed with a sharp cut-off on the lower side and a longer tail on its upper side. The secondary peaks seen in the 380 K and 400 K distribution result from the discrete cloud top values produced by the SAF-NWC retrieval. The intensity of this peak at 400 K is an indication that rare events of penetrative convection might make a significant contribution to the impact in the lower stratosphere. A better assessment of this penetrative convection is clearly needed to substantiate this finding. This is a challenge for present and future geostationary observations at high spatial and temporal resolution. The patterns of the horizontal distribution (panels e, h, k) are showing the same concentration of sources in North India and the Tibetan Plateau as for the forward case in Fig.4 with a decreasing contribution of over sources as the altitude rises. This localization of the convective sources for parcels reaching the top of the TTL was already observed, using backward trajectories, by Bergman et al. (2013) and attributed to the existence of a narrow conduit from the ground to the tropopause. We see here that the narrow conduit is that of convective towers bringing air from the boundary layer to the top of the clouds but that it stops there and that subsequent motion is distributed over a much wider area.

It is useful to compare the backward hit percentage and the forward impact. If we consider the two levels 380 K and 400 K, the areas of backward hit percentage larger than 70% for the first and 30% for the second are fairly similar, $15.210^6$ km$^2$ and $16.110^6$ km$^2$ respectively, and cover a large portion of the AMA domain, whatever definition is used. This shows obviously

that the convective influence propagates high in the lower stratosphere and that the in-mixing of background air is very limited. However, the ratios of the forward cumulated impact contained in these areas to that in the same areas at 355 K where it is

325 maximum can also be calculated and are only 12% and 4,5% respectively. There is no paradox, it only means that air escapes easily from the AMA as it ascends but penetrates much less easily inside. The distribution of backward hit percentage is also compatible with the observation of an apparent plume of tropospheric tracer in the AMA which has been reported in several previous studies based on large scale data and large-scale models (Park et al., 2009; Randel et al., 2010; Pan et al., 2016, e.g.). A columnar AMA rich in parcels influenced by convection and surrounded by background air is producing this very pattern.

Therefore we prove that observing such a tracer columnar pattern is not a proof of a chimney with impermeable walls as it is often assumed. It is generally known (see, e.g. Joseph and Legras, 2002) that forward trajectories link the stable structures at the initial point to the unstable structures at the final point in the future, while the backward trajectories link the stable structures at the initial point to the unstable structures at the final point in the past. Here the backward trajectories link the confined domain of the AMA to the LZRH surface, around which they tend to oscillate at long time while this same surface

repels forward trajectories.

## 3.4 Age

In forward runs, the age of air at a given level and within a given cell is defined as the mean over all the parcels contributing to the impact at this level within this cell. The same parcel can be counted several times with different ages since multiple crossings are possible. In backward runs, the age of air is calculated as the time interval between the parcel release on the

340 grid and the first hit of a convective cloud, and is averaged in the same way except that each parcel is counted only once. The forward age is shown in Fig. 10 for both the target and the sources. At 350 K, panel (c) shows a meridional split between air younger than one week in the South and older air in the North. The young region spans the easterly jet that carries rapidly the maritime air produced over the Bay of Bengal and the Sea of China to the West over Africa and outside of the FullAMA domain . At 350 K (panel c), the age is 2 days or less where the impact concentrates south of the Bay of Bengal at very close

proximity to the convective sources. The mean age is about 10 days over Sahara but much less over equatorial Africa. The age is larger in the AMA core region with a small impact due to the few parcels that recirculate within the AMA at this level. The pattern changes completely at 360 K (panel e) where a broad deep minimum age is seen at the core of the impact, surrounded by a region of larger age. The pattern persists at 380 K and 400 K (panels g and i) albeit with a reduced contrast between the core and the periphery. Therefore, the maximum impact density at the core of the AMA is maintained by the constant renewal

by fresh convective air rising from below rather than by the trapping. This provides support to the "blower" hypothesis of Pan et al. (2016). However, we find that the AMA blows parcels away uniformly from 360 K and up, and not only above the tropopause.

The distribution of age in the source space shows that at the lowest levels 340 K and 350 K (see Fig. 10(b,d)) the age is maximum over the continental regions and minimum over the maritime regions. The reason is that the continental air circulates

within the AMA before being expelled towards the major subsident regions over Arabia and Africa, while the maritime air is directly transported to these regions. At 360 K and above (see panels f,h,j), the mean age is fairly uniformly distributed

over the whole sources, indicating no preference for a faster path from any region. Sec. S8 of the Supplement shows that the age distribution of the backward trajectories agree very well with the forward distribution. Therefore we conclude that the age distribution indicates that the ascent occurs over a broad domain that covers the whole AMA rather than in a very localized region and that parcels remaining within the AMA spiral outward as they rise, as also found by Vogel et al. (2019).

### 3.5 Vertical crossover

In this section we concentrate on the domain where convection is the most active and relevant to the Asian monsoon. We divide this domain, labelled as Asia, into three subregions: Land, Ocean and Tibetan Plateau as shown in Fig. 11. The Tibetan Plateau is defined as the region with orography higher than 3800 m. We stress the need for a clear separation between ocean and land, that the commonly used rectangular boxes cannot provide. There are very noticeable differences, as shown in Tegtmeier et al. (2019), in reanalysis-based cloud properties and heating rates between the Bay of Bengal and the Sea of China on one side and the adjacent Indian subcontinent and South China on the other side.

Table 1 shows that the distribution of high clouds (as selected in Sec.2.4) favours the Ocean (68.4%) rather than the Land (26.6%) and the Tibetan Plateau (5%). The maritime convection is further divided (not shown in the table) among the Sea of China and the Philippine Sea (23%), the West and Mid Pacific (17,1%), the Bay of Bengal (14%) and the Indian Ocean (10,8%). The land is further divided among the Indian Subcontinent (12.8%), Indochina (7.8%) and South China (6.4%). The maximum high cloud cover is located 4 to 6 K higher on the Land than on the Ocean, and up to 10 K higher over the Tibetan Plateau. Table 1 also shows that the all LZRH is everywhere 5 K above the maximum high cloud cover in the ERA5 while in the ERA-Interim the two levels are close over Land and Ocean but separate by 8 K over the Tibetan Plateau.

For the three regions and Asia as a whole, and for all source levels, we determine the proportion of trajectories, rising, descending or being stationary. In this purpose we divide the potential temperature range [322.5 K, 422.5 K] into 20 bins of width 5 K and we calculate, for each region, a 2D histogram for the source level and the mean level of the convective parcels during their life time. In the FullAMA calculations, this latter is the period of time between the launch and the exit of the FullAMA box, the exit through the lower cut-off or the end of the integration. In the global case, the exit occurs only through the lower cut-off. We then calculate the rising proportion, for each level, as the proportion of parcels borne in this level for which the mean (life-time) level lies within bins above it. Similarly the descending proportion is calculated for parcels with mean level below the initial level and the stationary proportion corresponds to parcels with mean level within the initial bin. Figure 12, drawn for sources in the whole Asia region, shows a crossover at 362 K for both the FullAMA and global trajectories of EID. The crossover is located slightly above at 364 K for EAD, consistent with a smaller cloud radiative effect. At levels below the crossover, the two descent curves for EAD and EID within the FullAMA domain are very similar and drop rapidly due to the lower boundary. In the global domain, the drop is shifted to lower levels and delayed, due to the cross-hemispheric Hadley-Walker circulation. This separation between ascending and descending motion near 360 K was already mentioned by Garny and Randel (2016).

Table 1 shows the crossovers for the three component regions. In the ERA5, the crossover is above the all sky LZRH by 5.8 K over Ocean and 3.4 K over Land. It is below the all sky LZRH by 1 K over the Tibetan Plateau. The gaps are different

but the ordering is the same for the ERA-Interim. With respect to the clear sky LZRH, the crossover is only 1.2 K above over land but below by 5.2K over the Tibetan Plateau. In the ERA-Interim, the crossover is slightly below the clear sky LZRH over land. As the vicinity of the highest convective towers is filled by downdrafts we expect the crossover to be close but above the clear sky LZRH. The anomalous behaviour of the Tibetan Plateau with a crossover well below the clear sky LZRH but at the same level as the surrounding land can be explained if parcels detrained above the Tibetan Plateau leave rapidly the area by horizontal motion to find regions of ascent.

The fraction of the high clouds that are above the crossover and therefore contribute to the upward transport is quite low. It is minimum over Ocean, at 1.7% for ERA5. The Land value is more than twice larger (5.1%) and the Tibetan Plateau further doubles it (10.8%). As the ERA-Interim crossover is lower, the corresponding proportions are about twice that of ERA5. In both cases, continental convection is more likely than maritime convection to feed the upward motion above the LZRH and the Tibetan Plateau is by far the most efficient region as already found by Tissier and Legras (2016).

As a result of the crossover pattern, and because maritime convection is more easily washed outside the FullAMA domain, the relative contributions of Ocean and Land to the FullAMA impact at 380 K are inverted with respect to the proportions of high clouds in these regions (see Table 1). For the ERA5, the ratio Ocean/Land for the high cloud proportion is 2.57, and is 0.41 for the impact, therefore reduced by a factor 0.16. This is partially explained by the ratio of the high cloud crossover fractions, which is 0.33. The other explaining process is the washing out of the maritime impact. On the other side, the ratio Tibetan Plateau/Land is 0.19 for the high cloud and 0.41 for the impact. The enhancement by a ratio 2.2 is entirely explained by the high cloud crossover fraction ratio which is 2.1. Similar numbers can be derived for the ERA-Interim in the FullAMA domain. When we consider the global domain for the ERA-Interim, we see that the Ocean/Land impact ratio is 1.35, therefore a reduction by a ratio 0.52, slightly larger than the high cloud crossover fraction ratio 0.4.

These results show that the enhanced impact of the Tibetan Plateau is entirely due to the higher proportion of high clouds above the crossover in this region. The respective impact of oceanic versus land convection in the global domain is also mainly explained by this crossover proportion. There is more chance for parcels from continental convection to be trapped within the AMA but most of the air reaching the 380 K surface does not circulate first within the AMA. This result corroborates the findings of Tissier and Legras (2016) and Vogel et al. (2019).

## 4   A simple model of AMA confinement

In this section we investigate how the observed behaviour of the impact in the FullAMA region can be represented by a simple 1D model based on our observations. We consider a simple advection-diffusion model with loss for the impact $F(\theta, t)$:

$$\frac{\partial F}{\partial t} + \frac{\partial \dot{\theta} F}{\partial \theta} = -\frac{1}{\alpha}F + \kappa \frac{\partial^2 F}{\partial \theta^2} + S(\theta) \tag{1}$$

where $\alpha$ is the erosion time-scale, $\kappa$ is a vertical diffusion and $S(\theta)$ account for the distribution of high clouds.

The profile of the heating rate in the lower panel of Fig. 4 suggests that we can separate a region near the all sky LZRH $\theta_0$ where the heating rate grows linearly with $\theta$ and another region above where the heating rate is essentially constant. In

addition, as the LZRH lies above the level of maximum high cloud cover, the high clouds can be represented by an exponential distribution $S(\theta) = S_0 \exp(-\beta(\theta - \theta_0))$ with the slope found in Fig. 4.

In the simplest version, we assume that $\alpha^{-1} = 0$, therefore considering the global domain, and that the heating is $\dot{\theta} = \Lambda(\theta - \theta_0)$. In this case, (1) is a Fokker-Planck equation and it can be shown (Gardiner, 2009) that the probability of transit from $\theta_a$ to $\theta_b > \theta_a$ is

$$\Pi(\theta_a, \theta_b) = \frac{1 + \mathrm{erf}\left(\nu(\theta_a - \theta_0)\right)}{1 + \mathrm{erf}\left(\nu(\theta_b - \theta_0)\right)} \tag{2}$$

where $\nu = (2\kappa)^{-1/2}\Lambda^{1/2}$. Consequently, the distribution of convective sources that impact a given level is

$$P(\theta) = N^{-1}e^{-\beta(\theta - \theta_0)}\left(1 + \mathrm{erf}\left(\nu(\theta - \theta_0)\right)\right) \tag{3}$$

where the dependency on the impact level is in the constant $N$. The upper panel of Fig.13 shows that according to the value of the ratio $\beta/\nu$, the distribution of sources for parcels that travel to the target level $\theta$ is centred on the LZRH, below it or above it. Table 1 shows that over Land and Ocean, the proportion of sources above the LZRH is always large, up to 96% over the Ocean. Therefore we are in the case where $\beta > \nu$ and diffusion across the LZRH is negligible. The Tibetan Plateau differs by exhibiting

a majority of sources below the LZRH, especially for the ERA-Interim. This is consistent with the behaviour of the crossover and indicates that parcels rapidly travel outside the Tibetan Plateau, where the LZRH is lower, and they find ascending motion. On the overall, this suggests that the LZRH is an effective barrier and that diffusion due to fluctuating heating rates and explicit gravity waves resolved by the reanalyses is not large enough to overcome the barrier.

In the second stage, we neglect diffusion and we assume that $\dot{\theta} = \Lambda(\theta - \theta_0)$ from $\theta_0$ to $\theta_1$, above which the heating rate is

assumed to be constant equal to $A = \Lambda(\theta_1 - \theta_0)$. The LZRH is then a perfect barrier and we consider only the cloud sources above $\theta_0$ described with the same exponential distribution as in the previous stage. This problem can be fully solved (see Sec. S9 of the Supplement) and the solution is illustrated in the lower panel of Fig. 13, as a function of age at the level $\theta = 380$ K, for the basic set of parameters $\theta_0 = 360$ K, $\theta_1 = 370$ K, $\Lambda = 0.1$ day$^{-1}$, $\beta = 0.4$ K$^{-1}$, $\alpha = 10$ day and $A = 1$ K day$^{-1}$, that mimic the situation for the ERA5 diabatic transport in the monsoon area. The modal age and mean age are about 30 days

in rough agreement with Figs. S5 and S6 of the Supplement. Figure 13 also shows the effect of changing the parameters by multiplying $\Lambda$ by 3, $\beta$ by 0.4, $\alpha$ by 0.5 and $A$ by 3. It is visible that, for the displayed regime, $\Lambda$ basically controls the width of the distribution without changing the modal age. Changing $\alpha$ both changes the shape and the modal age. The parameter $\beta$, when reduced, leads to a new regime where the maximum is attained at $t = (\theta - \theta_1)/A$. When increased or less decreased (not shown), the effect is mainly a translation in time. $A$ performs essentially a translation in time. A full interactive solution as a

function of the parameters is available in the Supplementary material as a Computable Document Format notebook playable with Wolfram Player (https://www.wolfram.com/player/).

In the third stage, we calculate the solution by using the ERA5 and ERA-Interim heating rates within the restricted domain (20°E-140°E and 10°N-40°N) already used in Fig. 4 and the vertical distribution of the cloud sources used in the 3D calculations as shown in Fig. 4. The erosion time-scale $\alpha$ is fixed at 13.3 day and $\kappa = 0.1$ K$^2$day$^{-1}$ (that is about 0.05 m$^2$s$^{-1}$) is

used to regularize the solution. Figure 14 shows the distribution of impact as a function of age and potential temperature to

be compared with the right column of Fig. 5. The 1D model reproduces very well the main character of ascent and descent, albeit the temporal decay is faster than in the 3D calculations. A more quantitative comparison is made in Fig. 15 for ERA5 and the three isentropic levels 370 K, 380 K and 390 K, for several values of $\kappa$. We see that the diffusion basically slows the upward propagation but does not change qualitatively the solution. In view of its simplicity, the 1D model is very successful at reproducing the 3D solution, which means that it provides a relevant explanation of the mean transport and confinement properties of the AMA.

## 5    Conclusions

We have studied the transport pathways from injection at the top of the high convective clouds to the lower tropical stratosphere during the Asian monsoon, using a very dense set of Lagrangian trajectories driven by observed clouds and reanalysis data. We show that, unlike the ERA-Interim, kinematic and diabatic trajectories of the ERA5 provide a consistent description of the motion above the level of zero all sky radiative heating (LZRH). The kinematic and diabatic trajectories differ below the LZRH (missing in the kinematic case) within the convective region. However, both methods capture the descending motion over the deserts and the descending branches of the Hadley-Walker circulation.

The path of convective parcels depends on whether they are injected below or above the crossover level that separates mean ascending from mean descending trajectories. Below this level they are mostly entrained horizontally within the Hadley-Walker circulation towards regions of subsidence, where they return to lower levels. Above the crossover, parcels are entrained into the upward motion that lead them to cross the tropopause and enter the stratosphere. The crossover is at 364.4 K over Asia land with no significant difference over the Tibetan Plateau and is 2 K lower over the ocean. Due to the exponential decay of convective top frequency with altitude, only a small part of the convective clouds (2.6% on the average) reach high enough, above the crossover, to inject parcels that move further upward. The effective transport barrier of the crossover is usually located slightly above the clear sky LZRH. The Tibetan Plateau is an exception with a crossover lower than the all sky and clear sky LZRH but at the same level as surrounding land. This can be explained by the relatively small size of the plateau relative to the AMA and the ease of parcels to leave it by horizontal motion and subsequently find ascent motion in nearby regions of Asia.

In the region above the crossover, the apparent confinement within the Asian Monsoon Anticyclone (AMA) is the result of the constant renewal by fresh convective inflow and the leaky circulation of the AMA. As a result, the younger air is found at the core of the anticyclone and the oldest air is found at its periphery where it is expelled with a time-scale of about 13 days, which is of the same order but shorter than the returning time of the mean circulation (about 2 to 3 weeks). Therefore, the AMA is not a good container of the type of the stratospheric polar vortex. The sharpness of the boundary is however produced, like for the polar vortex, by the fact that expelled air is rapidly transported away and is equally rapidly replaced by exterior air in a mixng layer outside the boundary (see Joseph and Legras, 2002, for a detailed description of this mechanism). This erosion combines with a mean ascent rate of 1.1 $\mathrm{Kday}^{-1}$ to generate an exponential dilution with altitude. The dilution, partly due to the volume expansion, is compatible with the persistence of a strong convective influence within the AMA, through

the whole TTL as background air is mostly kept outside, thereby causing the appearance of a columnar tracer pattern found in many observations and modelling studies (Park et al., 2009; Randel et al., 2010; Pan et al., 2016). This process is akin to the "blower" hypothesis of Pan et al. (2016) except that it is uniformly distributed over the whole range of altitude from the crossover to 420 K at least. The ascent occurs therefore following a broad spiral, as advocated by Vogel et al. (2019).

The air that is found within the AMA comes mainly from continental convection. The sources exhibit a concentration in North India and the south of the Tibetan Plateau, as found in many previous studies (e.g., Bergman et al., 2013; Tissier and Legras, 2016). We find that, for continental convection as a whole, this result is partly due to the higher level of convection over land than over ocean. The localization of the convective sources and of the associated convective towers carrying air from the boundary layer was identified as a vertical "conduit" by Bergman et al. (2013). However, this conduit ends exactly at the level of cloud detrainment.

The Tibetan Plateau is favoured by its location at the core of the AMA and is also the region that exhibits the largest amount of high clouds above the crossover. This suggests that the compounds released at the ground there have the highest chance to reach the stratosphere. However, we find that the impact of the Tibetan Plateau at 380 K is entirely explained by the high proportion of clouds above the crossover. There is no indication of a favoured ascent above the Tibetan clouds. On the contrary, the fact that the crossover is lower than the local LZRH indicates that parcels leave the Tibetan Plateau to perform the ascent over other regions inside the AMA.

We find that the mean properties of upward transport and apparent confinement within the AMA over the whole summer can be explained by a simple 1D diffusive-advective-loss model, with a constant loss rate, forced by the observed distribution of convective sources and heating rates. A main ingredient to get an impact with both a maximum in age and altitude is that, between 360 K and 370 K, the heating grows from the all sky LZRH and that it stays roughly constant above 370 K up to the lower stratosphere. This is entirely consistent with the demonstrated broad and regular ascent in the AMA.

Our diagnostics are based on whole summer averages and ignore the variability during the season. Section S10 of the Supplement deals with this issue and shows that, at least in 2017, the pattern of the impact confinement does not change significantly over the whole summer, in spite of noticeable modulations in amplitudes and distribution within the AMA. The fact that the characteristic loss time is smaller than the circulation time indicates that the AMA confinement is fragile. It is modulated both by the source convective activity underneath, which is subject to a number of oscillations of the Monsoon, and by the modulation of filament shredding on the west and east edges which is also irregular, as discussed, e.g., in Pan et al. (2016) and Vogel et al. (2015). The coupling between these processes has been recently considered by Ortega et al. (2017) and Wei et al. (2019).

The forward trajectories ignore possible intersections with clouds after launch. Tissier and Legras (2016) showed that accounting such effect has a very small impact on the statistics of upward motion which is here our main focus. Our study is also limited by the quality of the observations and of the reanalyses. The estimation of high clouds from geostationary infrared imagers is subject to a number of uncertainties, in particular due to the cover of semi-transparent cirrus clouds above the anvils. The SAF-NWC algorithm detects such features but there are discontinuities in cloud classification between MSG1 and

Himawari-8 which have also a visible impact on the cloud height estimation. Combining imagers with sounders which are highly sensitive to ice clouds (Stubenrauch et al., 2013) will provide in the future a way to improve these retrievals.

Several recent studies (Hoffmann et al., 2019; Tegtmeier et al., 2020; Bucci et al., 2019) showed that the ERA5 improves the representation of atmospheric properties, including transport. The ERA5 is, however, singular in favouring very high penetrative convection over the Tibetan Plateau which should be considered with caution due to the lack of data and of training of the model over this region. The Tibetan Plateau, in spite of its limited global impact, is a region of high interest to understand transport within the AMA. The lack of high quality observations with active instruments, both from the ground and from space

(as current active instruments do not overpass the region in the evening when convection is the most active) hampers our understanding of convection over this region and certainly deserves some efforts to bridge the gap.

More generally, the fact that the clouds that contribute to the upward flux in the TTL and in the stratosphere are a small fraction within the upper tail distribution opens the question of the role of small-scale intermittent overshooting convection above the anvils which is seldom observed by the geostationary satellites. Although the effect was found by James et al. (2008)

to be small in the Asian Monsoon region, this deserves further investigation within the context of Lagrangian studies.

On the overall, our estimates of the convective impact using high resolution datasets and advanced satellite products essentially corroborate that of Tissier and Legras (2016), made with lower resolution data and less advanced estimates of the high clouds. It is also in good qualitative agreement with Garny and Randel (2016) regarding the role of the crossover and with Orbe et al. (2015) and Vogel et al. (2019) regarding the distribution of sources and provide new interpretation to the works of

540 Bergman et al. (2013) and Pan et al. (2016).

*Code and data availability.* Most of the programs used in this study are freely available and documented under github https://github.com/bernard-legras/STC/STC-forw and STC-back with dependencies in https://github.com/bernard-legras/STC/tree/master/pylib. This study generated a multi-terabyte ensemble of trajectories and post-processed files stored on the Institut Pierre-Simon Laplace meso-centre. They are all available from the main author upon request.

*Author contributions.* The design of the experiments, Lagrangian calculations and data processing have been performed by B.L. The analysis and the writing of the manuscript have been performed by B.L and S.B.

*Competing interests.* The authors declare no conflict of interest

*Acknowledgements.* We thank Alexandra Tzella for pointing out the Gardiner's solution to the exit time problem. This work was supported by the StratoClim project by the European Community Seventh Framework Programme (FP7/2007–2013) under grant agreement no. 603557, 550 and by the CEFIPRA 5607-1 and the TTL-Xing ANR-17-CE01-0015 projects. Meteorological analysis data were provided by the European

Centre for Medium-Range Weather Forecasts. ERA-5 trajectory computations were generated using Copernicus Climate Change Service Information. We also thank the AERIS Data and Service Centre for providing access to the MSG1 and Himawari data and to processing resources The Eumetsat SAF-NWC, and in particular Hervé le Gléau and Gaëlle Kerdraon, are thanked for giving access to their software

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

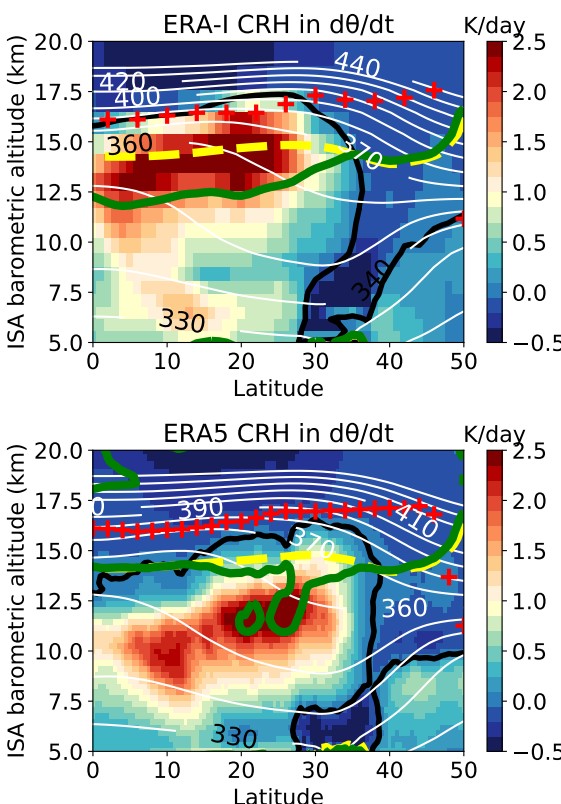

**Figure 1.** Cloud radiative heating (CRH) contribution to $d\theta/dt$ (in $\mathrm{K\,day^{-1}}$) averaged over July-August 2017 in the $73°\mathrm{E}$-$97°\mathrm{E}$ longitude range. Upper panel: ERA-Interim. Lower panel: ERA5. The black contour shows the zero line of CRH. White contours show potential temperature (in K). Red crosses show the cold point tropopause. The yellow line shows the clear sky LZRH. The green contour shows the all sky LZRH. The small green contour by 20 km above the equator in the ERA5 (masking the yellow line beneath) is associated with a weak descent in this region. The vertical axis is the barometric altitude derived from pressure using the hydrostatic equation and the standard atmosphere. The true geopotential altitude is higher, up to +850 m on the 360 K surface near 30N (see Fig. S1 of the Supplement).

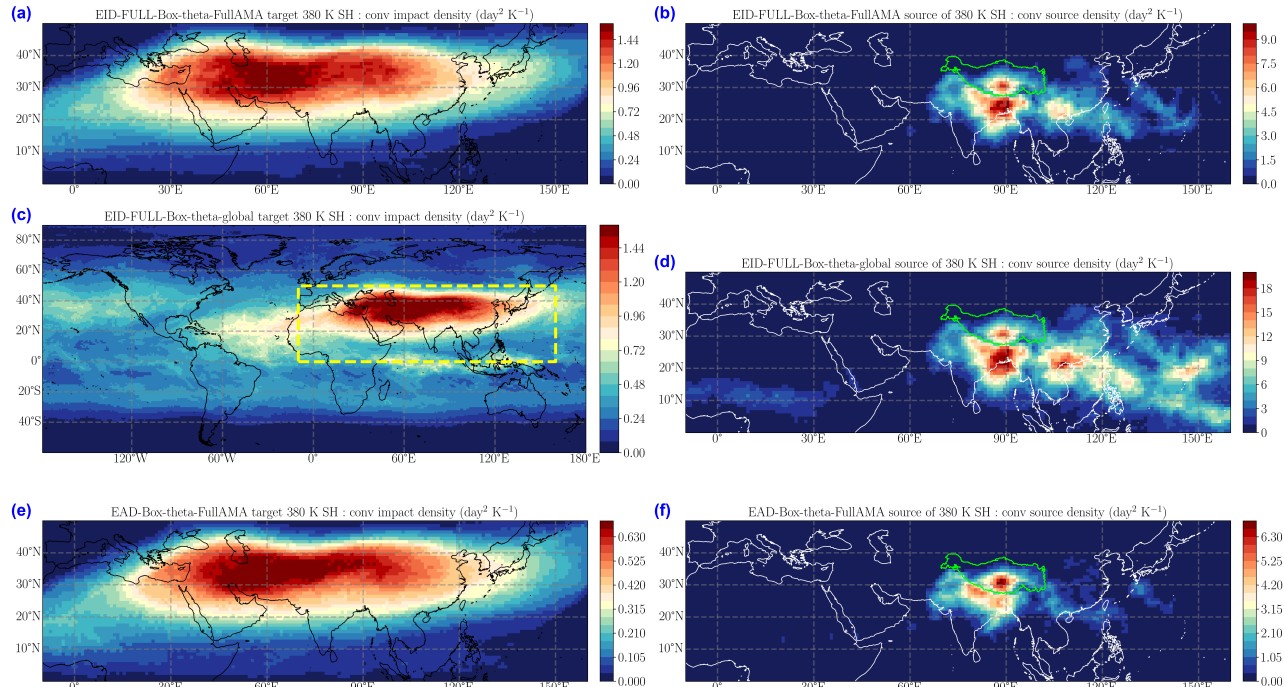

**Figure 2.** Impact density and source density for the convective parcels reaching the 380 K target level, accumulated over the 1st June - 31 August 2017 launch interval and the two-month life time of the parcels. Left column (a,c,d): the impact density at the 380 K target level. Right column (b,c,e): the source density of convective clouds from which the parcels reaching 380 K have been launched. Upper row (a,b): ERA-Interim diabatic trajectories in the FullAMA domain. Mid-row (c,d): ERA-Interim diabatic trajectories in the global domain. Lower row (e,f): ERA5 diabatic trajectories in the FullAMA domain. The green contour on the right panels shows the Tibetan Plateau.

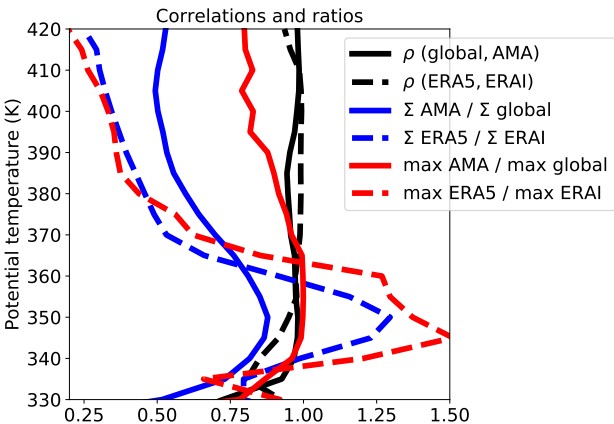

**Figure 3.** Solid back: correlation of the impact density between the FullAMA and global EID calculations within the FullAMA domain. Dash black: same for EAD and EID FullAMA calculations. Solid blue: ratio of maximum impacts between the FullAMA and global EID calculations within the FullAMA domain. Dash blue: same for EAD and EID FullAMA calculations. Solid red: ratio of cumulated impacts between the FullAMA and global EID calculations within the FullAMA domain. Dash red: same for EAD and EID FullAMA calculations.

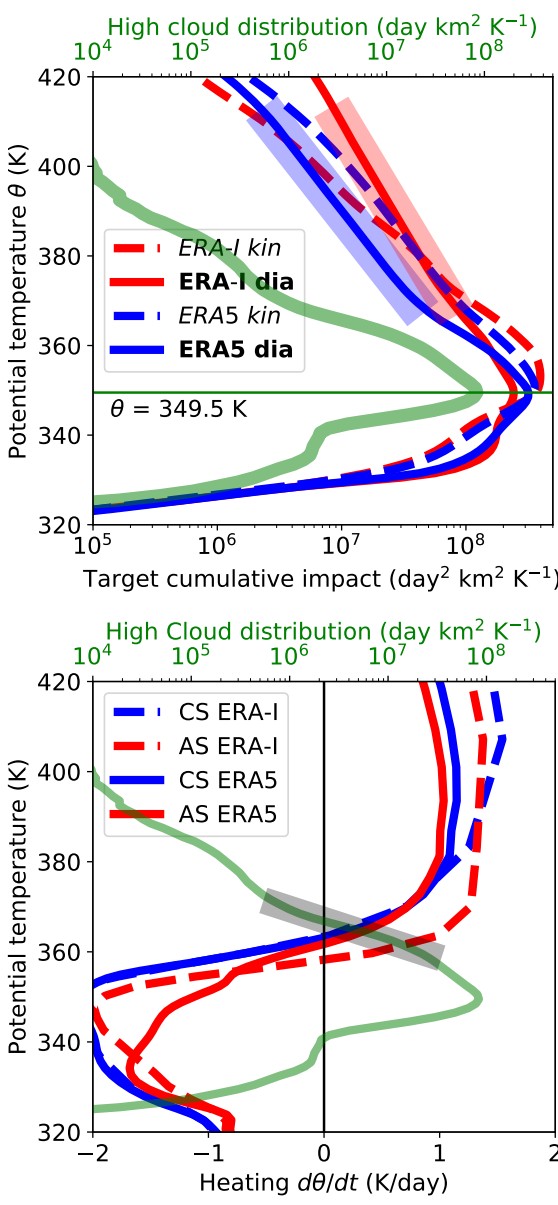

**Figure 4.** Upper panel. Green: Vertical distribution of the selected high cloud tops. Other curves: vertical distribution of the cumulated impact within the FullAMA domain for the ERA5 diabatic (EAD, solid blue), ERA5 kinematic (EAZ, dash blue), ERA-Interim diabatic (EID, solid red) and ERA-Interim kinematic (EIZ, dash red) experiments. For EAD and EID, a fit with a logarithmic decrement, respectively 0.065 K$^{-1}$ and 0.050 K$^{-1}$, is shown between 370 K and 400 K (wide segments).

Lower panel. Green: same as in upper panel. Other curves: radiative heating rate profile average over the restricted domain (20°E-140°E and 10°N-40°N) for July and August 2017 and all sky ERA5 (solid red), all sky ERA-Interim (dash red), clear sky ERA5 (solid blue) and clear sky ERA-Interim (dash blue). The fit of the high cloud distribution between 360 K and 370 K is shown with a logarithmic decrement $\beta = 0.325$ K$^{-1}$ (wide gray segment).

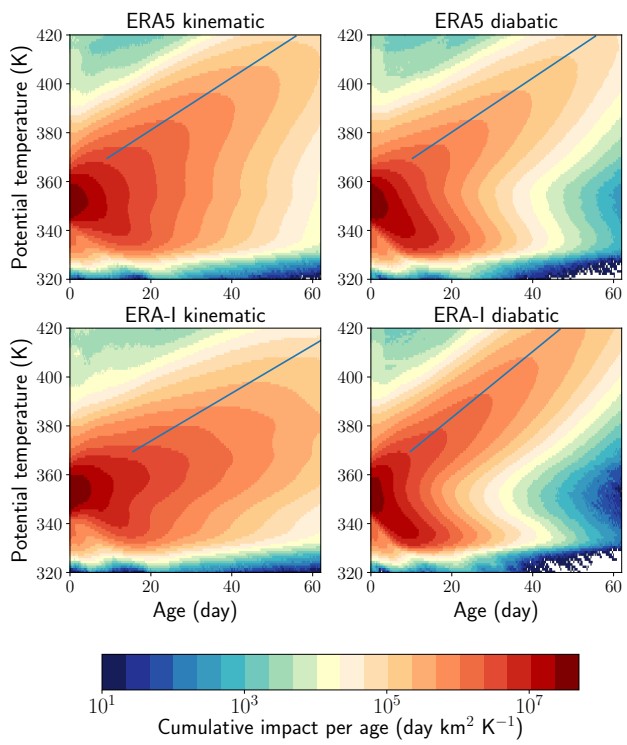

**Figure 5.** Distribution of the cumulated impact as a function of altitude and age with respect to the launch of the convective parcels. Upper row: ERA5. Lower row ERA-Interim. Left column: kinematic trajectories. Right column: diabatic trajectories. The slopes follow the crests of the distribution at each level (see also Sec. S4 of the Supplement).

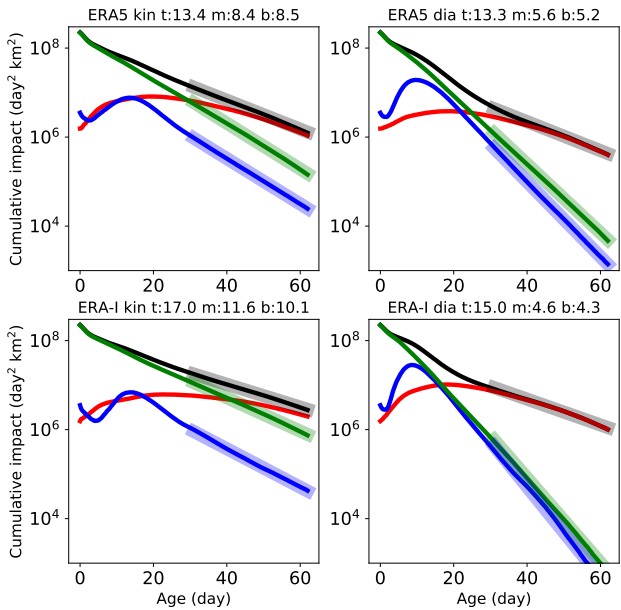

**Figure 6.** Evolution, as a function of age, of the total impact (black), the lower layer impact for $\theta < 340$ K (blue), the mid-layer impact for $340$ K $< \theta < 370$ K (green) and the upper layer impact for $370$ K $< \theta$ (red). The four panels are arranged as in Fig. 5. The asymptotic e-folding time of the total and upper layer for the four cases are 13.4, 13.3, 17.0 and 15 day for, respectively, EAZ (ERA5-kin), EAD (ERA5-dia), EIZ (ERA-I kin) and EID (ERA-I dia). The asymptotic e-folding times (in days) for the top-layer (t), mid-layer (m) and bottom-layer (b) are listed in the title of each panel. They are calculated from the fit of an exponential law between ages 30 days and 62 days shown as wide segments in the figure.

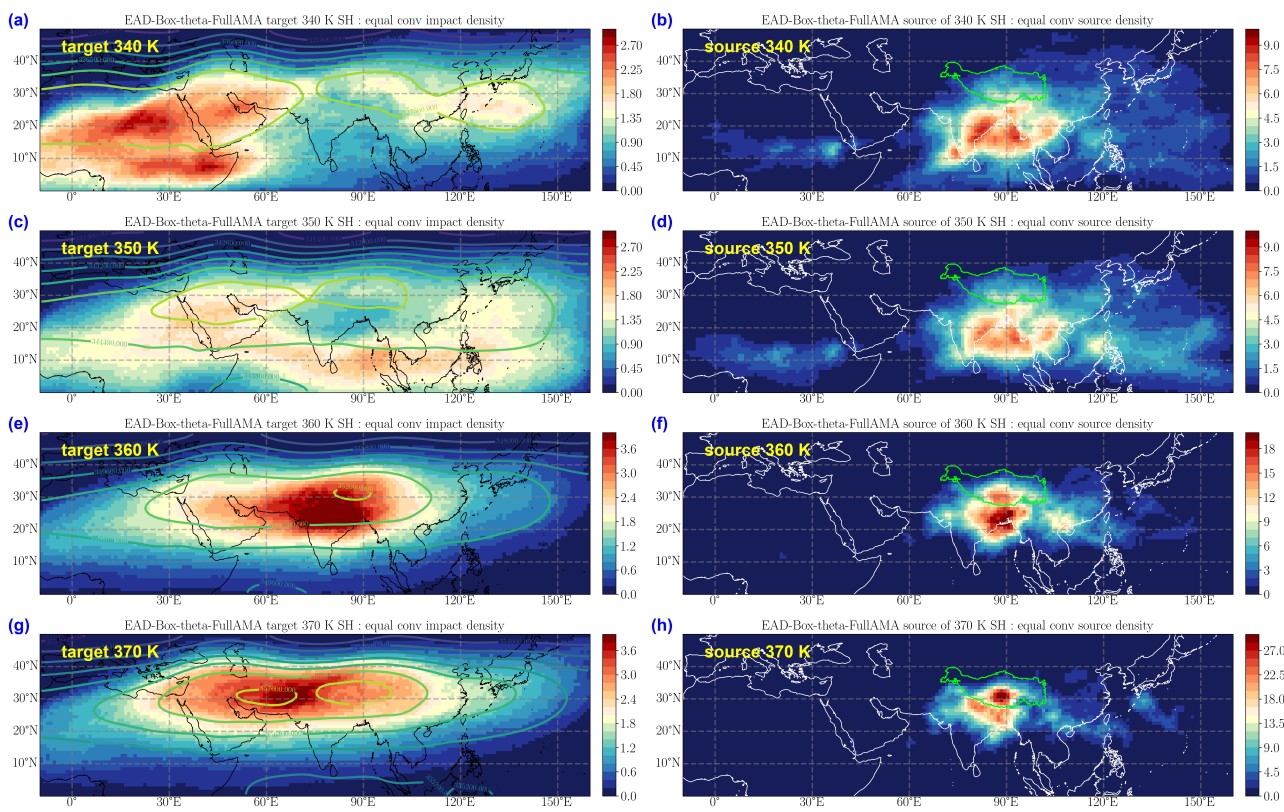

**Figure 7.** Left column (a,c,e,g): equalized impact density on isentropic levels (from top to bottom) (a) 340 K, (c) 350 K, (e) 360 K and (g) 370 K for ERA5 diabatic trajectories (EAD). Contours: Montgomery potential at the same levels. Right column (b,d,f,h): equalized source density for the same levels as in the left column and the same experiment.The green contour on the right panels shows the Tibetan Plateau.

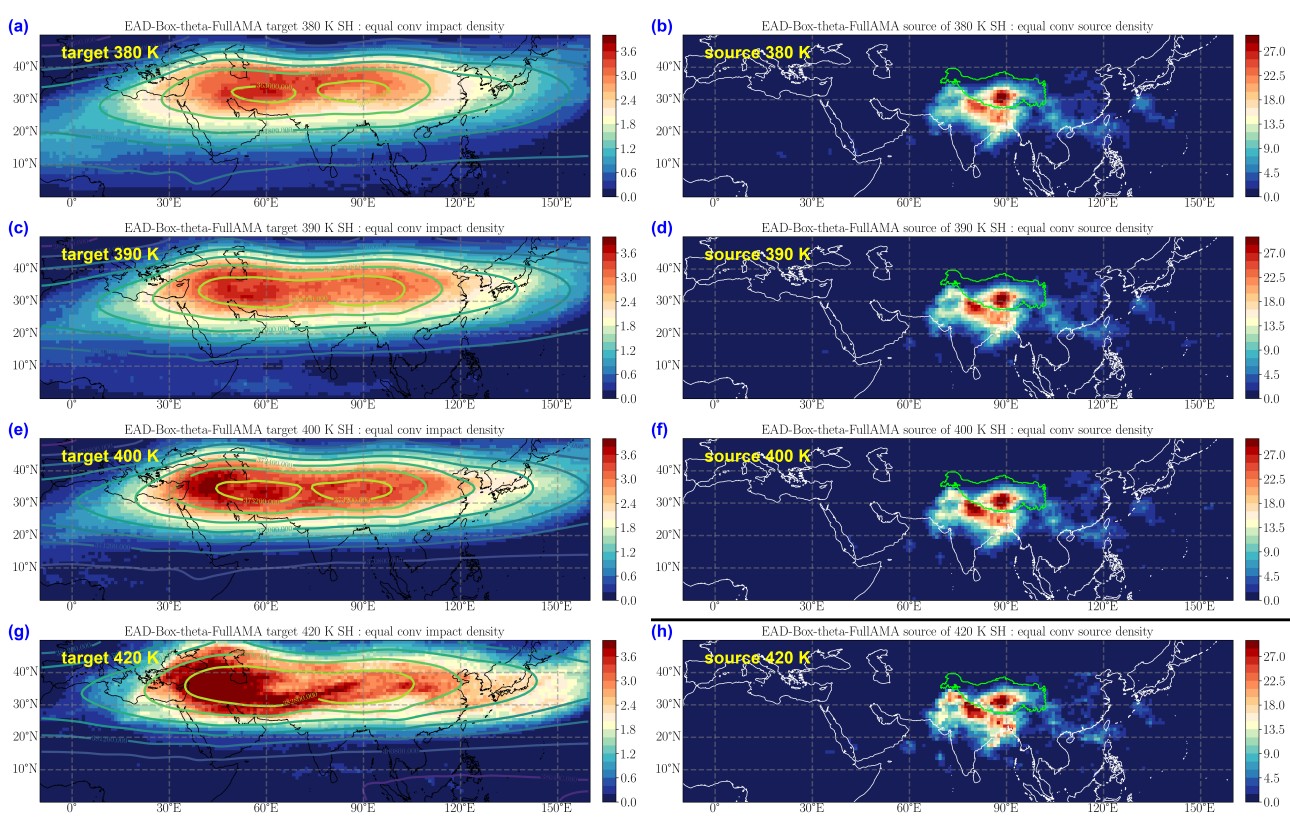

**Figure 8.** Same as Fig. 7 but for the levels (from top to bottom) 380 K, 390 K, 400 K and 420 K.

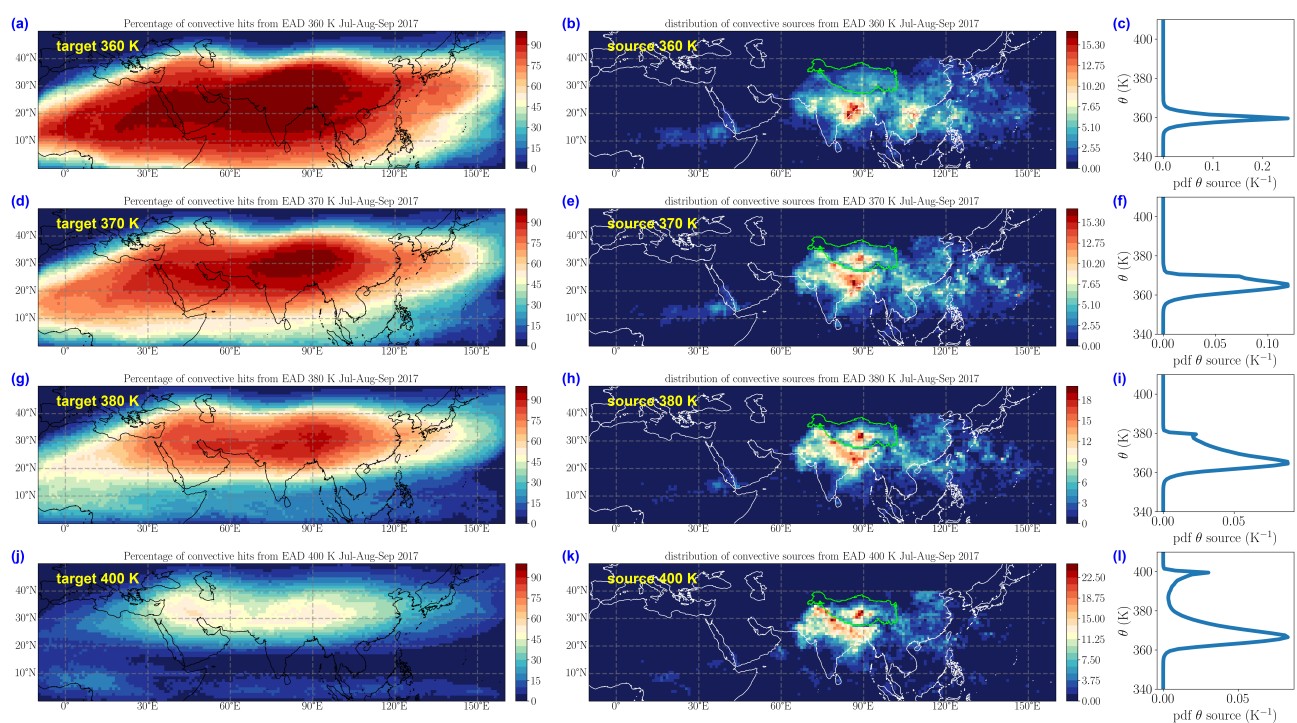

**Figure 9.** Left column (a,d,g,j): percentage of parcels hitting a convective cloud within 44 days for ERA5 diabatic backward trajectories starting at 380 K (a), 390 K (b), 400 K (g) and 420 K (j) over the interval 1 July - 30 September 2017. Centre column (b,e,h,k): equalized source density for the same levels as in the left column and the same experiment. Right column: vertical probability density function (in $K^{-1}$) of the cumulated source density for the same levels as the two left other columns.

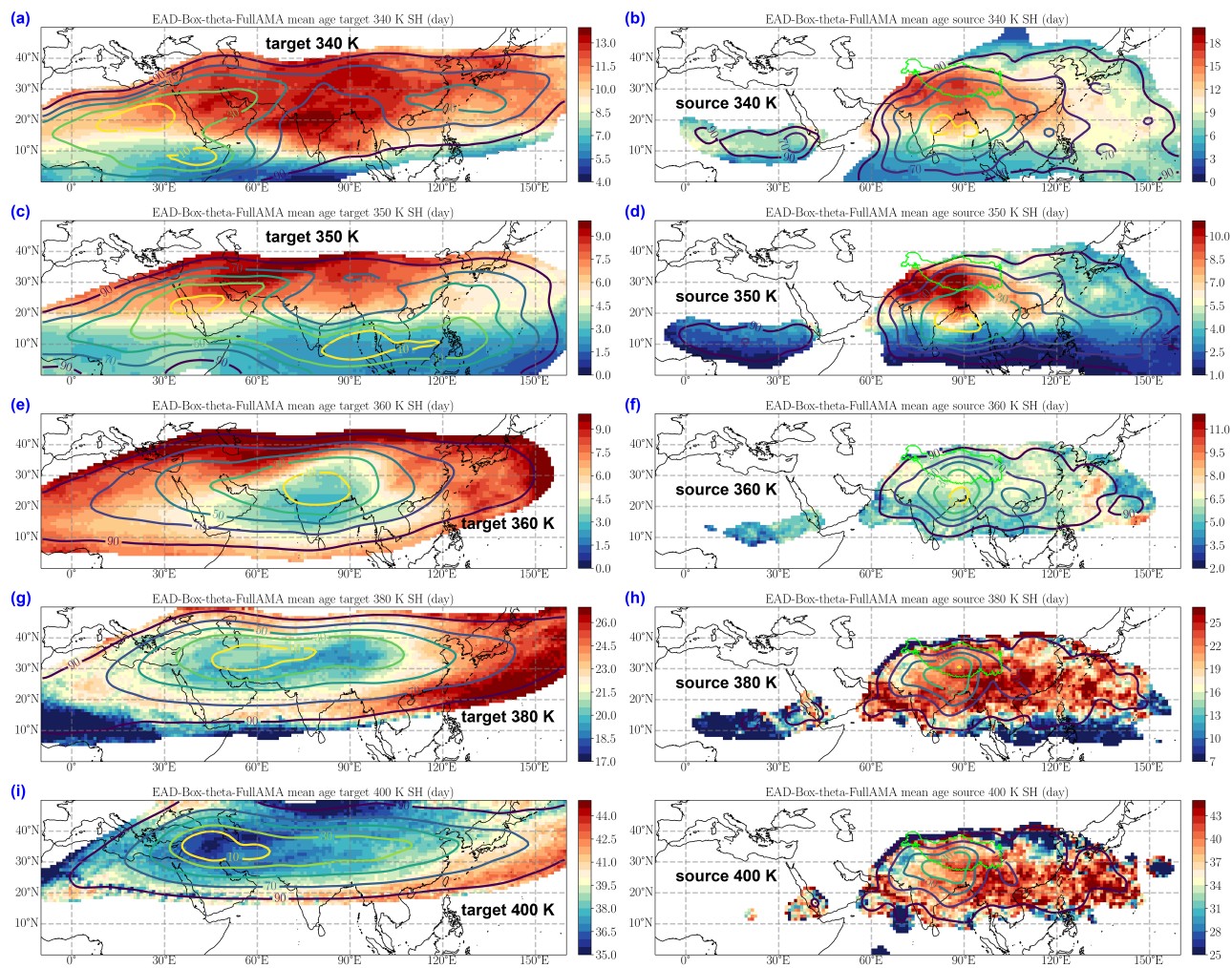

**Figure 10.** Left column (a,c,e,g,i): mean age (in day) with respect to convection for forward parcels reaching the levels 340 K (a), 350 K (c), 360 K (e), 380 K (g) and 400 K (i). Right column (b,d,f,h,j): mean age in the source domain for the same parcels as in the left column for each level. The contours show the impact distribution at the same level and the fields are clipped outside of the contour that contains 95% of the cumulated impact. The results are shown for the ERA5 forward diabatic trajectories (EAD)

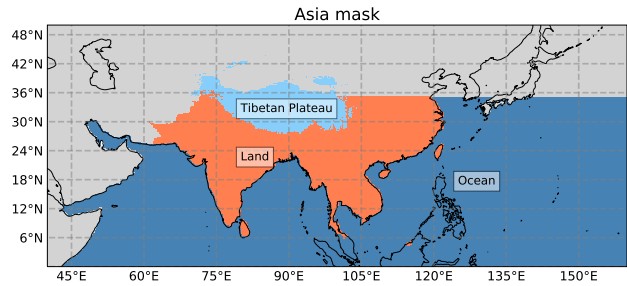

**Figure 11.** Mask of the three regions that partition Asia defined as their union.

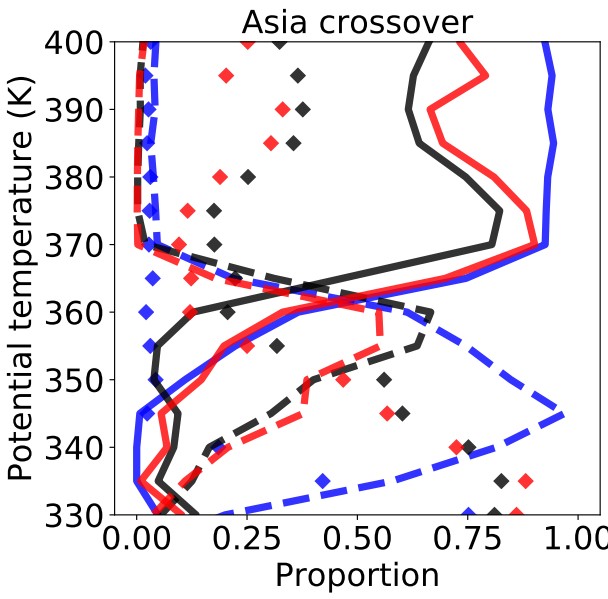

**Figure 12.** Solid and dashed curves: proportion of forward diabatic trajectories for which the mean vertical position during life-time is in a 5 K bin, respectively above or below, that of origin. Diamonds: proportion of forward trajectories with a mean vertical position within the origin 5 K bin. Black: ERA5 in the FullAMA domain; Red: ERA-Interim in the FullAMA domain. Blue: ERA-Interim in global domain. The curves are plotted for the whole Asia domain.

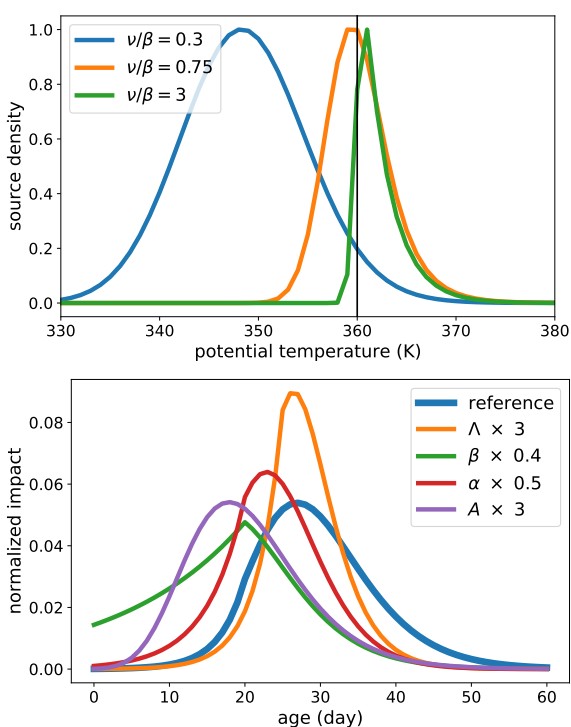

**Figure 13.** Upper panel: Distribution of convective sources $NP(\theta)$ according to (2) for $\theta_0 = 360$ K, $\beta = 0.4$ K$^{-1}$ and three values of $\nu/\beta$: 0.3, 0.75 and 3 as indicated in the legend. The proportion of sources above the LZRH $\theta_0$ in the three cases is, respectively, 4.7%, 55.8% and 97.4%. Lower panel: Solution of (1) described by (S5-S7) for the parameters given in the text and modified solutions according to the changes indicated in the legend. Each curve is normalized with respect to its integral in the displayed interval.

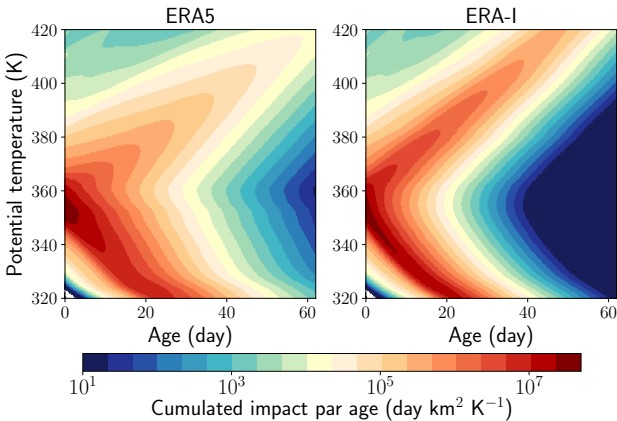

**Figure 14.** Solution of the equation (1) for the observed distribution of high clouds and heating rates within the restricted region defined in Fig. 4. Left panel for ERA5 is to be compared with the upper right panel of Fig. 5. Right panel for ERA-Interim is to be compared with the lower right panel of Fig. 5.

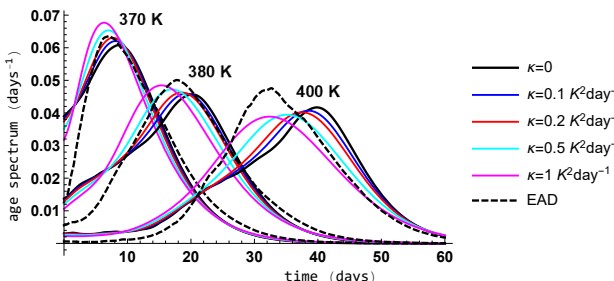

**Figure 15.** Solution of the equation (1) with the same set-up as in Fig. 14 for the diabatic ERA5 case. The normalized age spectrum is shown from left to right at 370 K, 380 K and 400 K for the inviscid solution ($kappa = 0$) and for four values of the diffusion $\kappa$: 0.1, 0.2, 0.5 and $1\,\mathrm{K}^2\mathrm{day}^{-1}$ as indicated in the legend. The normalized age spectrum for the 3D trajectories (EAD) is shown as a reference.

**Table 1.** Main characteristic numbers for the cloud distribution and the trajectories originating from Asia as a whole and its three sub-regions (Land, Ocean and Tibetan Plateau). When a proportion 100% is in the Asia column, the three other numbers in the row show the contributions of the three regions. EID is not separated into FullAMA and global cases when the distinction does not apply (LZRH and cloud fraction) or when it is negligible (crossover).

| | | Asia | Land | Ocean | Tibetan Plateau |
|---|---|---|---|---|---|
| High clouds SAF | Proportion | 100 % | 26.6% | 68.4% | 5.0% |
| | Max high cloud level | 349.5 K | 355.5 K | 349.5 K | 359.5 K |
| | Mean high cloud level | 352.9 K | 356.4 K | 351.1 K | 359.0 K |
| Clear sky LZRH | EAD | 358.9 K | 363.2 K | 357.0 K | 369.4 K |
| | EID | 359.3 K | 363.7 K | 357.5 K | 368.5 K |
| All sky LZRH | EAD | 357.9 K | 361.0 K | 356.7 K | 365.2 K |
| | EID | 352.9 K | 357.6 K | 351.0 K | 366.7 K |
| Crossover | EAD | 363.9 K | 364.4 K | 362.5 K | 364.2 K |
| | EID | 361.7 K | 361.8 K | 358.5 K | 363.1 K |
| High cloud fraction above crossover | EAD | 2.6% | 5.1% | 1.7% | 10.8% |
| | EID | 5.1% | 10.4% | 4.1% | 16.7% |
| Impact at 380 K and above | EAD FullAMA | 100% | 54.8% | 22.8% | 22.4% |
| | EID FullAMA | 100% | 54.4% | 32.0% | 13.6% |
| | EID global | 100% | 39.0% | 52.9% | 8.1% |
| Mean $\theta$ source for impact at 380 K and above | EAD | 366.0 K | 366.0 K | 367.2 K | 364.7 K |
| | EID FullAMA | 362.2 K | 362.7 K | 360.7 K | 364.1 K |
| | EID global | 359.2 K | 361.5 K | 356.7 K | 363.7 K |
| Proportion of source above LZRH for impact at 380 K and above | EAD | 95.2% | 83.5% | 95.6% | 36.5% |
| | EID FullAMA | 96.5% | 87.5% | 96.1% | 14.2% |
| | EID global | 94.1% | 74.1% | 81.3% | 12.5% |