# Peer review of "Confinement of air in the Asian monsoon anticyclone and pathways of convective air to the stratosphere during summer season"

_Atmospheric Chemistry and Physics, 2019_

## Referee Comment (RC1) · Anonymous Referee #1 · 13 Jan 2020

**General:**
This a very important and well-written paper and should be published by ACP. However, it is too long. It took me a few days to read and understand all details. I would recommend to include everything related to the "simple model" into a second paper. There is also some potential to shorten the remaining material (see below). There are still few major points which can certainly be clarified.

**Major points**

[Figure]

- "...The ERA5 diabatic and kinematic results differ not as much from each other than the respective results for ERA-Interim...". An important question is, if this is because the frequency of the data rather than data quality (less noise) became better for ERA5 if compared with ERA-Interim? Thus, if for ERA5 the data frequency would be reduced from 1 hour to 6 hour, would we also get similar large differences between the diabatic and kinematic calculations?

- You do not talk too much about the tropopause, especially as a vertical transport barrier during slow ascent of the spiraling trajectories. Because I do not see any effect of such a barrier in your results, I would recommend to mention this point in your discussion/conclusion or maybe even in the the abstract, i.e. something like: "tropopause is not a transport barrier for the vertically ascending trajectories..."

- This trajectory-based study with "true" convective clouds is very impressive and technically extremely well done. However, this is only an extremely model-based view of transport without any comparison with observations. Because of this, the picture of spiraling trajectories is a biased picture of reality (in the same way like "chimney", "conduit" or "blower"). Maybe the authors would like to re-consider this point.

- Horizontal confinement versus vertical transport. You derive some numbers showing that the horizontal confinement of the anticyclone is relatively weak (compared with the circulation rate). I am not convinced that the e-folding times derived from figure 7 really quantify the erosion rate, i.e., bi-directional crossing of air parcels across the vortex edge, a process that is typically believed to define the horizontal confinement (see also the minor comments).

- The 1d model is a great mathematical tool to put many things together and to create a simply 1d picture. But it is too much for this paper. Maybe to divide the whole material into two parts would help.

**Minor comments:**

- P1, L24
"due to a dominant radiative cooling" - not clear. Ascending motion means increase of the potential temperature of the considered air parcel. This means absorption of energy or warming. Maybe you should clarify this point.

- P2 L20-25
In your abstract you write: "We find no trace of a vertical conduit above convection over the Tibetan plateau": Maybe you should discuss in this paragraph also the concept of conduit and the related literature (mainly Bergman at al. papers). Furthermore, you discuss the concepts of "chimney" and "spiraling ascent" as two "extreme views". Please do not forget that the concept of "spiraling ascent" (which you seem to like more) is strongly related to long-term (40 days or more) and non-mixing trajectories. This is also a strong idealization, without any experimental confirmation in the observed tracers.

- P2 L34
"We also focus on reconciling previous studies...": This is for me too strong. The presented study including the 1D-model, does not contain any comparison with the experimental data. With the 1D-model you can get a different perspective on the presented, 3d-trajectory-based results. However, as mentioned above, I would recommend to have a separate paper describing the 1D model.

- P3 L21
Maybe you can finish the sentence in L21: "...at a $0.25^0$ horizontal resolution.", skip the lines 22 and 23 and follow with "The data are availably..." FullAMA will be explained in section 2.3.

- P3 L29
...cloud radiative heating (CRH) : Is it calculated as "all sky heating" - "clear sky

heating"?. It would be nice to mention it either in the text or in the caption of Figure 1. The green contour for ERA5 around the the equator between 17.5 and 20 km looks strange for me.

- P4 L10
  Are both terms: "latent heat" and "vertical heat diffusion" really available in both reanalysis products ERA-Interim and ERA-5?

- P4 L12-13
  I think, Ploeger et al., 2017 does not initialize the tracers in the boundary layer but above theta=360 K. Please reformulate.

- P4 L14
  ...converges rapidly to the (clear sky) radiative heating rate as a...

- P4 L19
  ...for several levels during summer 2017...

- P4 L19

- P5 L2
  S2 Montgomery potential. I did not find the derivation of (1) in Nakamura at al., 1995. Maybe this is the wrong citation or you should reference more exactly the used equation.

- P5 L 5
  Sattellitte - please correct

- P5 L 11
  in the FullAMA domain...?

- P5 L21
  "For each of these cloud pixels"...maybe better: "For each of the selected cloud pixels"

- P5 L27
  "...version is labeled EAZ. Additional two integrations..." - sounds better for me

- P5 L34
  "every 15' and 20": I would mention that this is the temporal resolution (see also L8)

- P6 L13-14
  It would be nice to know how the units of the convective impact factor shown e.g. in Fig. 2 (day$^2$/K) can be derived from your impact density

- P6 L18-20
  ...do not understand. sorry

- P6 L23
  "The impact can also be stratified according to": I am not sure that this is the best English formulation. Maybe "spectral-resolved with respect to age", however,"stratified" is certainly shorter

- P7, L2
  "...and decays from 0.88 at 350K to 0.52 at 420K"
  0.5 at 420K is really small. It means that only 50% of the convective sources in the FullAMA region are correctly counted. However, we are more interested in the air masses entering the stratosphere through the anticyclone and not through the FullAMA region. Maybe it would be better to calculate the accumulated impact for the AMACore rather than for the FullAMA region.
- Figure 3
  The used notation in this figure should be the same as in the main text (AMA → FullAMA, ("max global" and "max ERAI" is the same, maybe "max global (ERAI)", and the same with the $\sum$...)

- P 8 L 21
  ...Figure 4, bottom panel shows the distribution....

- P 8 L 22
  You can use your abbreviation AMACore...

- P 8 L 29
  ...propagates inside the FullAMA (?) domain from the sources...

- P 8 L 34
  Supplement: S3 is not used in the main text. Nevertheless it is interesting Maybe you can couple it with your main text.

- P 8 L 34, Supplement S4
  "The normalized impact then becomes an age spectrum...". In this definition of the age spectrum, mixing is not included. Although I know that this concept follows the idea of "irreducible air parcels" (e.g. Schoeberl et al., 2000), however, this is not the way how the age spectrum should be defined by integral operator replacing the full 3d advection-diffusion equation.

- P 9 L 8-9
  "As the vertical velocities are ascending everywhere" - All sky radiation in Fig. 4 shows negative values below 360 K both for ERA-I and ERA-5...? You mean the total diabatic heating rates and $w$ in the kinematic case which are shown in S5? "must is located" - must be located.

- Main text + supplement
  It seems that there are few good arguments supporting the kinematic approach to quantify the impact density at 340 and 350 K levels (descending branches of the Hadley and of the Hadley-Walker circulations). Maybe it is better to start your arguments with ERA-Interim (Figs. S9 and S10, where these branches are more visible) and continue with ERA-5 (Figs. S7 and S8). I also think that the partition of text (and arguments) between the main text and the supplement can be improved.

- P 9 L 17
  Once again, this is for me not the clean definition of the age spectrum (see above)

- P 9 L 19
  "The fastest propagation by EAD....while the slowest propagation by EID" - I think EAD and EID should be exchanged

- Figure 8
  The colors are not explained. Also it is not clear if the total impact is calculated for FullAMA or AMACore ?

- P 9 L 34
  What did you assume for $A$? According to your equation, $A = 1/(0.69\alpha)$, it will lead to $A = 0.109$ K/day. I do not see in Figure 4b that this is the mean value of the diabatic ascent. On the other hand, you use in your 1d model values of $A = 1$ K/day. Please clarify

- P 9 L 34
  I am also not really convinced that $\alpha = 13.3$ day derived from Figure 7 can be understood as the erosion rate. Clear, the number of convective air parcels is becoming smaller with the altitude because, as you show, the total convective impact decreases with the altitude. But it is not clear, if these air parcels stay e.g.

at the (inner) edge of the anticyclone (with low diabatic heating rates) or really cross this edge and move, roughly isentropically, into the regions outside of the anticyclone (such crossing of the edge could be interpreted as the erosion of the anticyclone).

- P 9 L 34
  If your calculation is correct, you get a picture of "upward spiraling air within a chimney" crossing the tropopause and maybe crossing the edge of the anticyclone (see above). The latter behavior, above the tropopause, is sometimes denoted as a blower.

- P 10 L6-24
  This part of the text is very similar to the discussion on page 9, L8-15. The figures 8 and 9 are almost the same like figures S7 and S8 discussed before. I think, the logic can be improved here. An idea for the sensitivity study would be to check how your results (total impact) will change if you confine your region from FullAMA to AMACore defined by different values of the Montgomery stream function.

- Figure 10
  The characters of the potential temperature values in the last column are too small I also think that this figure can be moved to the supplement.

- P 11 L4
  ...contribution of sources as the altitude rises...

- General
  In all your pictures it is difficult to localize the Tibetan Plateau.

- Figure 11
  Left column: This is a very nice way to show how the spiraling stars in the center

of the anticyclone (or spiraling within the chimney). Right column: why you do not use the same color bars like in the left column. I would expect the same spatial distribution of the mean age like in the left column. Because of this, figures (h) and (j) look very confusing for me.

- P 11 L32
  maybe: ...in reanalysis-based cloud properties and...

- P 12 L30-37
  Here I lost your way of arguments. Maybe a stronger relation to the lines in the table and more small-steps explanation of the numbers (like 2.57, 0.41 or "inverted contribution") would help

- P 13 L 13
  You should give here the definition of $S = S_0 \exp{-\beta}$....

- P 13 L 23
  "Consequently, the distribution of convective sources..." I thought that this distribution is prescribed by $S$. Maybe you mean the distribution of air parcels originating from the convective sources

- P 14 L 5
  I think, the figure S12 is so important to understand the 1d-model that it cannot be moved to the supplement. By the way, I did not find at which level all the distributions are shown in figure S12.

- P 14 L 5
  Parameter $\alpha$ describing the erosion rate. It is not clear if this parameter in the 1d-model is the same like that introduced in 3.2. The numbers seem to be very different. In relation to the 1d-model you do not introduce it in the main text and even in the supplement you do not say what is the basic value of $\alpha$ (S9 L8-10).

- P 14 L 19
  "...provide a consistent description.." - see my first main comment

- P 14 L 31
  "...leaky circulation..." - I do not see a good proof of this statement (see my comments above)

- P 15 L 1-2
  "...As the level rises, the confined Asian monsoon is more and more diluted..." - same type of problem. See also abstract: "The contrast is reduced by dilution...". Which type of dilution: in-mixing of old stratospheric air into the anticyclone? But you do not have any proof of that.

- P 15 L 18
  "...rather than concentrated in a narrow pipe." - It means for me, it is certainly not a conduit but much more a chimney, with some upward spiraling air inside.

---

## Referee Comment (RC2) · Anonymous Referee #2 · 22 Jan 2020

This study utilizes a Lagrangian-based approach to quantify the influence of convection on transport to the upper-level monsoon anticyclone. The explicit use of observed clouds, in combination with new proposed measures of "convective influence", is a welcome contribution to the growing number of studies that have been oriented around better understanding the influence of convection on transport into the monsoon UTLS region. While various interesting results are presented I am concerned with the lack of investigation surrounding potential sensitivities of the authors' calculations (e.g. to trajectory length). The overall presentation is also a bit sloppy which renders it difficult to extract the main key messages from the study. Finally, the incorporation of an idealized model at the end is a great addition but better suited for development in a

separate study that can afford the space. As such I recommend acceptance subject to major revisions that address the following concerns:

Major Comments:

1) There is an overall sloppiness in the layout and presentation. This is especially the case for the writing and overall construction of the various sections and, to a lesser extent, with the figures. In particular, in addition to various typos, the authors repeatedly use single one-sentence paragraphs that break the continuity of the flow and reflect an overall lack of attention to the overall structure of the manuscript. See lines 18-28 on page 6, lines 21-25 on page 7, lines 14-15 on page 9, lines 18-20 on page 10, lines 26-27 on page 20, etc. My concern with this is not just one of aesthetics – rather, I think it reflects a lack of overall coherence so that the paper reads more like a point-by-point description of an exercise that was implemented and less like a coherent story that ties together different results in a clear, consistent manner. The problem this creates is that it makes it difficult for the reader to extract the main messages of the text. Please revisit these sections and try to present more cogently.

2) While I find it admirable that the authors are presenting a range of new diagnostics for quantifying the convective impact on various transport measures I am concerned that little discussion is presented as to how these measures depend on, among others, the time over which the trajectories are evaluated (i.e. the backward trajectories are followed for two months since their initial launch). The transport measures inferred from trajectories are notoriously sensitive to how long they are followed and I am surprised that this is not discussed. Perhaps these tests have been performed in previous studies but no mention is made in the current text. A discussion along these lines (and including other potential sensitivities) should ideally be presented either in the Methods section or in the Conclusions in order to communicate to the reader which measures are more robust, compared to others.

On a related point, I am discouraged by the lack of details in some of the measures

presented. In particular, the concept of the "age" is introduced rather haphazardly on line 23 on page 6 and the details at the beginning of Section 3.4 do little to place this definition of an age in the context of previous studies. This would not be an issue so much if the authors did not then proceed on page 9 to claim "Note that the commonly used metric of the mean age might hinder...". In particular, the mean age as defined here is based on a spectrum of ages that are only evaluated for 2 months of the integration. The mean age, therefore, as defined here ignores the influence of longer timescales that may substantially influence the tails of the age spectrum (e.g. Hall and Plumb (1994)) and, thereby, skew the mean to higher ages. Indeed, this is quite evident in the 400K "age spectra" shown in the black lines in Figure 6 where the pdfs of the ages have clearly not converged within the 60 days considered in the study. Numerous previous studies have shown that Eulerian measures of the mean age are the averages of very broad underlying age spectra (e.g. Waugh and Hall (2002)) and that not considering these longer timescales (which physically measure the contribution of older recirculating air parcels) can have a significant impact on the mean age. If the readers do not plan to address this explicitly in their analysis (which would require extending their trajectories) then they should at the very least be very clear that what they mean by the "mean age" is very different than the mean age as defined in previous studies.

3) The use of the cloud information from SAF-NWC is clearly a great contribution afforded by this study. I am concerned, however, about the potential dynamical inconsistencies that are introduced through use of observed cloud fields with large-scale meteorological variables derived from the reanalyses and, moreover, what these inconsistencies imply for transport. Given that the resolved vertical velocities used in the kinematic calculations will be largely dictated by the horizontal resolution of the underlying reanalysis model it is not obvious apriori that the observed clouds will be at all similar to the assumed (parameterized) clouds in the reanalysis models. This potential for inconsistency should be at least discussed in the manuscript.

4) The addition of the simple model in Section 4 is nice. However, given the already

extensive scope of the study which, upon the improvements suggested in my previous comments would add still more length and detail, I feel that Section 4 is too much. It would be more appropriate to include in another study. This would afford the authors more latitude and space to explore various sensitivities in their age and other transport measures evaluated here. The latter would really help convince the reader about the robustness of their results.

Technical Comments:

General: Please reduce the number of acronyms used throughout as this contributes to the difficulty in following the text. Some acronyms are really not appropriate as they are not standard in the field (e.g. EID and EIZ, FullAMA) whereas others are, clearly, more appropriate (AMA) as they have precedent in the literature.

page 1 line 2: "massive" is a strange word to use here. Is this really necessary? page 1 line 2: reanalysis -> reanalyses. This applies throughout the manuscript. page 2 line 1: leave room "for" not "to" page 2 line 7: "repelling" is a strange word as it does not imply with respect to what (i.e. is a dynamical or transport barrier?) – I suggest removing. page 2 line 19: "a dispersion" -> This term is too physical. Perhaps "lack of consensus"? page 3 line 17: So, which winds are actually used here? Analysis or forecast? page 3 line 18: What is meant by "interspelling". I strongly suggest using a more common term. I also don't understand this sentence. Are the heating rates coincident in time with the winds? Fig. 7: What are blue/red/green lines? Labels are not explained in the caption.

---

## Author Comment (AC1) · 19 May 2020

**Answer to reviewer #1**

B. Legras and S. Bucci

**Correspondence:** bernard.legras@lmd.ens.fr

We thank the reviewer #1 for his/her thorough discussion that led to significant improvements of our manuscript and a reinforcement of our conclusions.

As Fig. 6 has been removed from the paper, a shift in the numbering of the subsequent figures occurs in the new version. We therefore refer to these figures with the new number followed by the old in parenthesis.

5 **Answer to general comments**

– *"...The ERA5 diabatic and kinematic results differ not as much from each other than the respective results for ERA-Interim...". An important question is, if this is because the frequency of the data rather than data quality (less noise) became better for ERA5 if compared with ERA-Interim? Thus, if for ERA5 the data frequency would be reduced from 1 hour to 6 hour, would we also get similar large differences between the diabatic and kinematic calculations?*

10 First, the ERA-Interim winds are used at 3-hourly resolution because we interleave 3h and 9h forecasts between the analysis, at 0h and 6h, for each 12h assimilation cycle. The heating rates which are a product of the forecast are also used at 3-hourly resolution. The effect of using hourly, 3-hourly and 6-hourly data in the ECMWF operational model for transport studies with TRACZILLA was studied by Pisso et al. (2010) who concluded that the improvement was important from 6-hourly to 3-hourly but was modest from 3-hourly to hourly. On the contrary, Angevine et al. (2014) 15 found than hourly wind were improving the results of FLEXPART but this was for a mesoscale application.

It would take many efforts to redo our calculations at degraded resolution but one can compare the variances of the heating rates and the velocities shown in Fig. A1-1 in the core of the AMA. The ERA5 hourly variance is much larger than the 3-hourly ERA-Interim variance in the whole TTL for the vertical velocity and up to than 380 K for the heating rate. The variance of ERA5 3-hourly data calculated as averages of the hourly data is, by construction, smaller than 20 that of the hourly data. However, it is much larger than the minimum value that would obtain if the hourly data were decorrelated from one time to the next. This means that a significant correlation of hourly heating rates and vertical velocities persists over three hours and that we do not expect a very significant change by degrading the resolution to 3h. The differences in transport properties with the ERA-Interim are too large to be explained this way.

[Figure]

**Figure A1-1.** Mean profiles (left column) and variances (right column) for the all sky heating rate (2 upper rows) and the vertical velocity (lower row). Blue curves are for the ERA-interim data at 3-hourly resolution, red curves are for the ERA5 data at hourly resolution and black curves are for the averaged 3-hourly ERA5 data. The dashed black curve shows the 3-hourly ERA5 variance under the hypothesis of uncorrelated hourly data which is one-third of the hourly variance. The calculations are performed in the domain ($20°$E-$140°$E and $10°$N-$40°$N) over July and August 2017.

– *You do not talk too much about the tropopause, especially as a vertical transport barrier during slow ascent of the spiraling trajectories. Because I do not see any effect of such a barrier in your results, I would recommend to mention this point in your discussion/conclusion or maybe even in the the abstract, i.e. something like: "tropopause is not a transport barrier for the vertically ascending trajectories..."*

We thank the reviewer for raising this point. It is true that the tropopause does not seem to play any particular role in slowing the transport or changing its character. The mean ascent rate of $1.1 \ \mathrm{Kday^{-1}}$ extends smoothly through it. The important barrier level is the crossover level that one can see, within the diabatic framework, as the Lagrangian mean extension of the zero level of radiative heating rate.

– *This trajectory-based study with "true" convective clouds is very impressive and technically extremely well done. However, this is only an extremely model-based view of transport without any comparison with observations. Because of*

*this, the picture of spiraling trajectories is a biased picture of reality (in the same way like "chimney", "conduit" or "blower"). Maybe the authors would like to re-consider this point.*

Our approach has been extensively tested against observations up to 400 K in a companion paper (Bucci et al., 2019) where we used the same tools in a comparison of Lagrangian reconstructions with data from the StratoClim campaign. We found that the ERA5 diabatic trajectories provided the best interpretation of the data. Our methods have also identified the sources of ammonium measured at high altitude by following back trajectories spiralling in the AMA to the agriculture sources (Höpfner et al., 2019) and further checks will appear in other works to be published. The robustness of the modelling approach is therefore supported by observations.

– *Horizontal confinement versus vertical transport. You derive some numbers showing that the horizontal confinement of the anticyclone is relatively weak (compared with the circulation rate). I am not convinced that the e-floding times derived from figure 7 really quantity the erosion rate, i.e., bi-directional crossing of air parcels across the vortex edge, a process which is typically believed to define the horizontal confinement (see also the minor comments).*

The erosion time scale 13.3 day is actually quite comparable to the circulation time which is about two weeks. Figure 6(7) is based on the loss rate of the FullAMA box. Therefore, parcels accumulating at the periphery of the anticyclone would not be lost. Figure 8(9) shows that from 370 K to 420 K, the impact pattern remains quasi-constant but for the exponential decay shown in Fig. 6(7). There is no accumulation on the edge or elsewhere that builds up. A fit to the exponential decay of the total impact with altitude between 370 K and 400 K gives a slope of 0.065 $K^{-1}$ which is to be compared to the inverse of the product between the ascent rate 1.1 $Kday^{-1}$, that is 0.068 $K^{-1}$. Therefore, all the pieces fit together and the exponential rate found in Fig. 6(7) is a good representation of the erosion rate.

– *The 1d model is a great mathematical tool to put many things together and to create a simply 1d picture. But it is too much for this paper. Maybe to divide the whole material into two parts would help.*

We consider that the 1-D model is an important part of the paper as it fulfils the demonstration that the average transport properties of the Asian Monsoon Anticyclone can be quantitatively reduced to a small set of simple parameters. Without that part, we think that the paper will be considerably weakened. We notice that the reviewer considers below that one of the figure produced from the 1-D model is so important that it should be moved from the supplement to the main text.

**Answer to minor comments**

Some of these comments might have deserved to be in the major section but we follow the ordering of the reviewer.

– *P1, L24 "due to a dominant radiative cooling" - not clear. Ascending motion means increase of the potential temperature of the considered air parcel. This means absorption of energy or warming. Maybe you should clarify this point.*

The main detrainment of the convective clouds occurs at a level below the level of zero radiative heating (LZRH) where the air is almost everywhere descending. This is accompanied by radiative cooling. It has been often questioned

how this air could find its way above the LZRH (Corti et al., 2006). Our answer is that it does not. The LZRH or its Lagrangian version we call the crossover is a barrier and only clouds detraining above can feed the mass flux that enters the stratosphere.

- *P2 L20-25 In your abstract you write: "We find no trace of a vertical conduit above convection over the Tibetan plateau": Maybe you should discuss in this paragraph also the concept of conduit and the related literature (mainly Bergman at al. papers). Furthermore, you discuss the concepts of "chimney" and "spiraling ascent" as two "extreme views". Please do not forget that the concept of "spiraling ascent" (which you seem to like more) is strongly related to long-term (40 days or more) and non- mixing trajectories. This is also a strong idealization, without any experimental confirmation in the observed tracers.*

This is an important point that has led us to add some text in Sec. 3.3 and to rephrase a few sentences in the conclusions. We essentially recover the results of Bergman et al. (2013) in the compact distribution of sources shown in Fig. 8(9) and even more explicitly in Fig. 10(11) that shows the sources of backward trajectories similar to those they calculated. In our set-up, the high convective sources are assumed to loft parcels directly from the boundary layer located about 15 km beneath. This is what accounts of the conduit found in Bergman et al. (2013). The main interest of our work is that we separate the fast vertical transport that occurs in the cloud and the much slower transport of detrained parcels that occurs outside the convective towers. It is this second part which is the focus of this work. It is much more difficult to make such separation when doing calculations based on reanalysis winds down to the surface since the vertical velocities in the convective layers are averages of upward and downward motion at the mesh size. These velocities do not represent the speed of the transport of the surface to the TTL which takes a few weeks instead of a few tens of minutes. This smooth transport does not see the clouds and cannot say where the conduit ends. Our answer is that it ends at the detrainment level of the clouds. The clouds that provide air entering the stratosphere are those that reach above the crossover which is quite uniform over whole Asia. This crossover is located slightly above the continental LZRH and this is expected from the 1D model if diffusion due to heating variance does not play a significant role. The Tibetan plateau is the only region where the crossover is below the LZRH. This means that the parcels need to leave the Tibetan plateau to rise to higher altitude. This is what they do following the spiralling paths we have identified. This spiralling motion is not an idealization or a numerical fantasy, there are plenty of confirmations by observations in the analysis of the StratoClim data (see Höpfner et al., 2019; Bucci et al., 2019). it is the very essence of motion within the AMA.

On the other side, our results are also compatible with the findings of Ploeger et al. (2017) who launched forward trajectories near 360 K. The impact pattern being almost constant with height preserves a maximum at the core of AMA at high altitude as noticed by Ploeger et al. (2017). However, this does not imply per se that transport is more vertical than horizontal across a given surface like the tropopause. This property depends a lot on the slope of this surface with respect to the isentropes. Our analysis which is conducted from the potential temperature perspective does not exhibit any discontinuity in the range of levels spanned by the tropopause. Our simple 1D view of constant ascent and constant is, on the seasonal average, essentially valid from the crossover at 364 K to 420 K.

See also our discussion of the backward versus forward trajectories below.

- *P2 L34 "We also focus on reconciling previous studies...": This is for me too strong. The presented study including the 1D-model, does not contain any comparison with the experimental data. With the 1D-model you can get a different perspective on the presented, 3d-trajectory-based results. However, as mentioned above, I would recommend to have a separate paper describing the 1D model.*

  Again, look at Bucci et al. (2019) regarding direct comparisons with tracer data. Our claim is that the main features of the mean 3D transport can be represented quantitatively in terms of a few parameters and that this is quantitatively equivalent to a simple 1D model. Physics is full of other examples where a mean field theory describes the mean behaviour of an apparently very complex system.

  Our discussion is able to include the findings from previous studies and to provide a new interpretation for some of them that eliminates the apparent contradictions. For instance, we show that the existence of a columnar tracer pattern within the AMA, cannot be taken as a proof that the AMA behaves as a closed "chimney" with no leak as it has been often assumed on the basis of a qualitative interpretation. We show that the chimney hypothesis is contradicted by the detailed analysis of the transport properties and that instead the columnar tracer pattern is compatible with the broad ascent with uniform loss that is the product of the transport. Observations are not limited to tracers but include temperatures and winds which are assimilated to produce the velocity fields. There is no doubt that state of the art NWP systems produce analysed horizontal winds which are a good approximation of the true circulation. Our work is investigating the consequences of these observations within the well validated framework of geophysical fluid dynamics. It is therefore incorrect to say that we do not take account of experimental data. We actually take account of both tracer and dynamical observations and we arrive at a new interpretation that is not contradicted by any of them. This is the way in which we reconcile the previous studies which did not adopt this global perspective.

- *P3 L21 Maybe you can finish the sentence in L21: "...at a 0.25 0 horizontal resolution.", skip the lines 22 and 23 and follow with "The data are availably..." FullAMA will be explained in section 2.3.*

  Good suggestion. We have done that.

- *P3 L29 ...cloud radiative heating (CRH) : Is it calculated as "all sky heating" - "clear sky heating"?. It would be nice to mention it either in the text or in the caption of Figure 1. The green contour for ERA5 around the the equator between 17.5 and 20 km looks strange for me.*

  The cloud radiative heating is calculated in that way, now mentioned in the text. The green contour at the equator above the tropopause comes from the data. It corresponds to a descent in the stratosphere by 20 km above the equator that is modulated by the QBO. This feature exhibits some slight variability among the reanalyses.

- *P4 L10 Are both terms: "latent heat" and "vertical heat diffusion" really available in both reanalysis products ERA-Interim and ERA-5?*

No, there is a total temperature tendency and there are all sky and clear sky long wave and short wave heating rates. The residual, latent heating + diffusion, can be obtained by subtracting the all sky heating rate from the total. There is also a separate heat coefficient diffusion which might help to reconstruct the vertical heat diffusion. We have changed the sentence to avoid suggesting the two quantities are separately available.

– *P4 L12-13 I think, Ploeger et al., 2017 does not initialize the tracers in the boundary layer but above theta=360 K. Please reformulate.*

Corrected.

– *P4 L14 ...converges rapidly to the (clear sky) radiative heating rate as a...*

It is not only clear sky. Radiative heating by clouds cannot be neglected even high above their top. For instance high opaque clouds reduce significantly the upward long wave flux in the whole column above their top. It was proposed by Corti et al. (2006) that the warming produced by overlaying cirrus helps parcels to cross the LZRH. What we neglect here is the cooling effect due to the evaporation of falling ice in clear air.

– *P4 L19 ...for several levels during summer 2017...*

Corrected

– *P5 L2 S2 Montgomery potential. I did not find the derivation of (1) in Nakamura at al., 1995. Maybe this is the wrong citation or you should reference more exactly the used equation.*

The derivation is not in Nakamura (1995) but it can be easily derived using the framework of this work. We provide now a more detailed derivation as this relation might be of some practical usage.

– *P5 L 5 Sattellitte - please correct*

Correction done

– *P5 L 11 in the FullAMA domain...?*

Correction done.

– *P5 L21 "For each of these cloud pixels"...maybe better: "For each of the selected cloud pixels"*

Correction done

– *P5 L27 "...version is labeled EAZ. Additional two integrations..." - sounds better for me*

Correction done

– *P5 L34 "every 15' and 20": I would mention that this is the temporal resolution (see also L8)*

Correction done

– *P6 L13-14 It would be nice to know how the units of the convective impact factor shown e.g. in Fig. 2 (day 2 /K) can be derived from your impact density*

There was a typo error in the normalizing formula given in Sec. 2.5. $\Delta\theta$ should not appear with an exponent 2 in the denominator. So $\mathrm{day}^2\mathrm{K}^{-1}$ is the correct unit for impact density. Therefore the cumulated age is using $\mathrm{day}^2\mathrm{km}^2\mathrm{K}^{-1}$ units and the age histogram (in Fig.5) is using $\mathrm{day}\,\mathrm{km}^2\mathrm{K}^{-1}$ units. Thanks for spotting that. The units used in the figures were correct. We also use the equalized impact density which is without unit.

– *P6 L18-20 ...do not understand. sorry*

These lines have been rewritten.

– *P6 L23 "The impact can also be stratified according to": I am not sure that this is the best English formulation. Maybe "spectral-resolved with respect to age", how- ever, "stratified" is certainly shorter*

This sentence has been removed as the age is only needed in Sec. 3.4. We now prefer "sorted" to "stratified".

– *P7, L2 "...and decays from 0.88 at 350K to 0.52 at 420K" 0.5 at 420K is really small. It means that only 50the FullAMA region are correctly counted. However, we are more interested in the air masses entering the stratosphere through the anticyclone and not through the FullAMA region. Maybe it would be better to calculate the accumulated impact for the AMACore rather than for the FullAMA region.*

At 420 K, the dilution is of a factor 26 with respect to the level 370 K and the air composition is dominated by the stratospheric background (as confirmed by unpublished observations of the StratoClim campaign). What we count are the parcels in the FullAMA domain which never leave the domain. The many parcels that leave the domain below 420 K still climb to higher levels in the outside world and can later re-enter the FullAMA domain after completing a large loop or, most often, a full latitude circle. This might be a debatable issue but we tend to consider these parcels as not different from the background and no longer as fresh convective particles as they have been transported over long distances in filamentary structures that have been strained and stretched a lot resulting in mixing with surrounding air. It is nevertheless possible that a significant convective influence is preserved at 420 K within the AMA like at 400 K (as shown from backward calculations).

The FullAMA domain has been chosen in such a way that almost all parcels leaving the anticyclone are also leaving the domain soon after. This was done in order to avoid resorting to an a priori definition of the AMA boundary which presents considerable difficulties on a day to day basis. The AMACore domain remains most of the time inside the AMA but there is no reason to assume that the parcels spend most of there time in this domain. Figure 10(11) actually shows they tend to be ejected to the periphery.

– *Figure 3 The used notation in this figure should be the same as in the main text (AMA → FullAMA, ("max global" P and "max ERAI" is the same, maybe "max global (ERAI)", and the same with the ...)*

We agree that it was quite difficult to follow this discussion. Fig. 3 is now better referred in the text.

- *P 8 L 21 ...Figure 4, bottom panel shows the distribution....*

  Corrected, this was a relic of a previous different layout.

- *P 8 L 22 You can use your abbreviation AMACore...*

  We have removed this acronym as it was used only a couple of times and therefore was no necessary.

- *P 8 L 29 ...propagates inside the FullAMA (?) domain from the sources...*

  Corrected

- *P 8 L 34 Supplement: S3 is not used in the main text. Nevertheless it is interesting Maybe you can couple it with your main text.*

  It is indirectly used in Sect. 3.2 when we refer to Sect. S2 of the Supplement.

- *P 8 L 34, Supplement S4 "The normalized impact then becomes an age spectrum...". In this definition of the age spectrum, mixing is not included. Although I know that this concept follows the idea of "irreducible air parcels" (e.g. Schoeberl et al., 2000), however, this is not the way how the age spectrum should be defined by integral operator replacing the full 3d advection-diffusion equation.*

  Some sort of mixing is actually present here under the form of time averaging statistics which is equivalent to real mixing under a statistically stationary assumption and our approach could be given an integral representation. Irreducible air parcels are most often used to solve problems which have also an integral representation (see, e.g., Legras et al., 2005). Our definition of the age spectrum does not differ from the common usage.

- *P 9 L 8-9 "As the vertical velocities are ascending everywhere" - All sky radiation in Fig. 4 shows negative values below 360 K both for ERA-I and ERA-5...? You mean the total diabatic heating rates and w in the kinematic case which are shown in S5? "must is located" - must be located.*

  Vertical velocities mean Dz/dt, nothing to do with heating. This paragraph has been rewritten and is hopefully more readable. When total diabatic heating, including latent heating, is included, an ascending motion results everywhere in the monsoon region. We have done calculations with the total heating version of the ERA5. Above the crossover the results are almost the same as those obtained with the radiative heating only, so all the conclusions of our work are unchanged. Below the crossover the results are closer to that of the kinematic calculations. We considered that this was sort of expected (and reassuring) and decided it was not necessary to include these results.

- *It seems that there are few good arguments supporting the kinematic approach to quantify the impact density at 340 and 350 K levels (descending branches of the Hadley and of the Hadley-Walker circulations). Maybe it is better to start your arguments with ERA-Interim (Figs. S9 and S10, where these branches are more visible) and continue with ERA-5 (Figs. S7 and S8). I also think that the partition of text (and arguments) between the main text and the supplement can be improved.*

This would lead to a long discussion but we do not see why the kinematic approach is inadequate to describe the descent over the subsident regions of the Hadley-Walker circulation which are essentially free of clouds. The kinematic calculation misses the descent in the monsoon region but this is not the main part of the returning flow and we cannot say that our radiative calculations are very accurate in the range of altitudes below the anvils. In this domain, the local potential temperature profiles undergo frequent inversions due to convective mixing combined with horizontal transport and we neglect cooling by evaporation of stratiform precipitations. Generally speaking, potential temperature is hardly usable as a vertical coordinate in the convective layers of the atmosphere.

– *P 9 L 17 Once again, this is for me not the clean definition of the age spectrum (see above)*

See answer above.

– *P 9 L 19 "The fastest propagation by EAD....while the slowest propagation by EID" - I think EAD and EID should be exchanged*

There was an inversion but Fig. 6 has been removed as it brought nothing necessary and could be miss interpreted.

– *Figure 8 The colors are not explained. Also it is not clear if the total impact is calculated for FullAMA or AMACore ?*

This refers probably to Fig. 6(7) where the indication of the colours was indeed missing in the caption. The total impact is calculated in the FullAMA domain.

– *P 9 L 34 What did you assume for A? According to your equation, A = 1/(0.69α), it will lead to A = 0.109 K/day. I do not see in Figure 4b that this is the mean value of the diabatic ascent. On the other hand, you use in your 1d model values of A = 1 K/day. Please clarify*

There was a decimal missing in both numbers (0.069 and 0.065). Thank you for spotting that. $A$ is not derived from this formula but from the mean ascent rate in Fig. 5. The inverse of the product $A\alpha$ is compared to the impact slope in the upper panel of Fig. 4.

– *P 9 L 34 I am also not really convinced that α = 13.3 day derived from Figure 7 can be understood as the erosion rate. Clear, the number of convective air parcels is becoming smaller with the altitude because, as you show, the total convective impact decreases with the altitude. But it is not clear, if these air parcels stay e.g. at the (inner) edge of the anticyclone (with low diabatic heating rates) or really cross this edge and move, roughly isentropically, into the regions outside of the anticyclone (such crossing of the edge could be interpreted as the erosion of the anticyclone).*

This question is already asked in the main comment on the erosion rate and answered above. The parcels need to cross the edge as they cannot leave the FullAMA domain otherwise. If parcels are removed from FullAMA domain, they are a fortiori removed from the AMA. There is no other way.

– *P 9 L 34 If your calculation is correct, you get a picture of "upward spiraling air within a chimney" crossing the tropopause and maybe crossing the edge of the anticy- clone (see above). The latter behaviour, above the tropopause, is sometimes de- noted as a blower.*

The erosion rate is quite uniform above 370 K. There is no indication that the parcels stay inside the AMA seen as a wide chimney and that they are blown away only when they reach sufficiently high. The notion of a blower is correct if it is not localized only above the tropopause but uniformly distributed in potential temperature above the crossover. The idea of a chimney is incorrect as it implies that nothing leaks outside its wall (otherwise it cannot be considered as a good chimney).

– *P 10 L6-24 This part of the text is very similar to the discussion on page 9, L8-15. The figures 8 and 9 are almost the same like figures S7 and S8 discussed before. I think, the logic can be improved here. An idea for the sensitivity study would be to check how your results (total impact) will change if you confine your region from FullAMA to AMACore defined by different values of the Montgomery stream function.*

Actually, we refer to figures S7 and S9 of the supplement but only to provide supplementary information with respect to the discussion of figure 8 regarding the descent branches and the global versus FullAMA description. The core of the discussion is, however, contained in the main text. We do not see in which way Figs. 7(8) and 8(9) are almost the same. Fig. 7(8) shows the descending branch and the crossover and Fig. 8(9) shows the similarity of the ascending branch.

– *Figure 10 The characters of the potential temperature values in the last column are too small I also think that this figure can be moved to the supplement.*

The right panels of the figure have been improved. We still think that this figure is useful as the design of the backward calculations is quite similar of that of Bergman et al. (2013) and allows to interpret their finding of a vertical "conduit". This figure also shows the in-mixing of background air (loss of backward trajectories to the background) as a function of altitude and is critical to answer one of the questions of the reviewer below.

– *P 11 L4 ...contribution of sources as the altitude rises...*

Modified

– *General In all your pictures it is difficult to localize the Tibetan Plateau.*

We have added a contour for the Tibetan plateau on all the panels describing the sources.

– *Figure 11 Left column: This is a very nice way to show how the spiraling stars in the center of the anticyclone (or spiraling within the chimney). Right column: why you do not use the same color bars like in the left column. I would expect the same spatial distribution of the mean age like in the left column. Because of this, figures (h) and (j) look very confusing for me.*

Although the range of ages should not be that different, the weighting is very different in the target and the source space and there is no correspondence between cells with a given value in the target space and cells with the same value in the source space. Same remark holds for the impact shown in Figs. 7(8) and 8(9). If there was a preferred conduit from the sources to the 380 K level or the tropopause, this would shown up as a minimum age in the AMA core. This is not the case and we must reject the hypothesis of such a preferred conduit.

– *P 11 L32 maybe: ...in reanalysis-based cloud properties and...*

Modification done.

– *P 12 L30-37 Here I lost your way of arguments. Maybe a stronger relation to the lines in the table and more small-steps explanation of the numbers (like 2.57, 0.41 or "inverted contribution") would help*

We have rewritten and hopefully improved the description and the discussion of the crossover which is a central result of our work.

– *P 13 L 13 You should give here the definition of $S = S_0 exp - \beta$ ...*

Actually, it has to be given a bit below as $S(\theta)$ in (1) can also be the observed distribution from the SAF-NWC products.

– *P 13 L 23 "Consequently, the distribution of convective sources..." I thought that this distribution is prescribed by S. Maybe you mean the distribution of air parcels originating from the convective sources.*

This was confusing and has been corrected. We mean the distribution of convective parcels that reach a given level. The factors depending on this level are contained in the constant $N$.

– *P 14 L 5 I think, the figure S12 is so important to understand the 1d-model that it cannot be moved to the supplement. By the way, I did not find at which level all the distributions are shown in figure S12.*

We followed the advice of the reviewer and moved this figure to the main text, merging it with Fig. 13(14) so that the number of figures does not increase. The figure are plotted at $\theta = 380$ K.

– *P 14 L 5 Parameter $\alpha$ describing the erosion rate. It is not clear if this parameter in the 1d-model is the same like that introduced in 3.2. The numbers seem to be very different. In relation to the 1d-model you do not introduce it in the main text and even in the supplement you do not say what is the basic value of $\alpha$ (S9 L8-10).*

The basis parameters at the second stage of the 1D model include $\alpha = 10$ day as only the order of magnitude is needed to show the sensitivity to variations of parameters. In the third stage which is directly compared to the 3D trajectories, we use $\alpha = 13.3$ day that is the value derived in Sec.3.2. This should be now clear in the text.

– *P 14 L 19 "...provide a consistent description.." - see my first main comment.*

Please refer to our answer above.

– *P 14 L 31 "...leaky circulation..." - I do not see a good proof of this statement (see my comments above)*

Please refer to our answer above.

– *P 15 L 1-2 "...As the level rises, the confined Asian monsoon is more and more diluted..." - same type of problem. See also abstract: "The contrast is reduced by dilution...". Which type of dilution: in-mixing of old stratospheric air into the anticyclone? But you do not have any proof of that.*

The in-mixing from the background is demonstrated in the backward calculations (as an increasing amount of backward trajectories come from the background with increasing height). This is consistent with all the other results of our study.

It is useful to compare the backward hit percentage and the forward impact. If we consider the two levels 380 K and 400 K in Fig. 9(10), the areas of backward hit percentage larger than 70% for the first and 30% for the second are fairly similar, $15.210^6$ km$^2$ and $16.110^6$ km$^2$ respectively, and cover a large portion of the AMA domain, whatever definition is used. This shows obviously that the convective influence propagates high in the lower stratosphere and that the in-mixing of background air is very limited. However, the ratios of the forward cumulated impact contained in these areas to that in the same areas at 355 K where it is maximum can also be calculated and are only 12% and 4,5 respectively. There is no paradox, it only means that air escapes easily from the AMA as it ascends but penetrates much less easily inside. The distribution of backward hit percentage is also compatible with the observation of an apparent plume of tropospheric tracer in the AMA which has been reported in several previous studies based on large scale data and large-scale models (Park et al., 2009; Randel et al., 2010; Pan et al., 2016, e.g.). A columnar AMA rich in parcels influenced by convection and surrounded by background air is producing this very pattern. Therefore we prove that observing such a tracer columnar pattern is not a proof of a chimney with impermeable walls as it is often assumed. It is generally known (see, e.g. Joseph and Legras, 2002) that forward trajectories link the stable structures at the initial point to the unstable structures at the final point in the future, while the backward trajectories link the stable structures at the initial point to the unstable structures at the final point in the past. Here the backward trajectories link the confined domain of the AMA to the LZRH surface, around which they tend to oscillate at long time while this same surface repels forward trajectories. This paragraph has been added to Sec. 3.3

– *P 15 L 18 "...rather than concentrated in a narrow pipe." - It means for me, it is certainly not a conduit but much more a chimney, with some upward spiraling air inside.*

A chimney with walls at the edge of the AMA is excluded by our analysis as discussed in the previous comments and we demonstrate that the observation of a columnar pattern of tropospheric tracer within the AMA cannot be taken as a proof of a chimney.

**References**

Angevine, W. M., Brioude, J., McKeen, S., and Holloway, J. S.: Uncertainty in Lagrangian Pollutant Transport Simulations Due to Meteorological Uncertainty from a Mesoscale WRF Ensemble, Geoscientific Model Development, 7, 2817–2829, https://doi.org/10.5194/gmd-7-2817-2014, 2014.

5 Bergman, J. W., Fierli, F., Jensen, E. J., Honomichl, S., and Pan, L. L.: Boundary Layer Sources for the Asian Anticyclone: Regional Contributions to a Vertical Conduit, Journal of Geophysical Research: Atmospheres, 118, 2560–2575, https://doi.org/10.1002/jgrd.50142, 2013.

Bucci, S., Legras, B., Sellitto, P., D'Amato, F., Viciani, S., Montori, A., Chiarugi, A., Ravegnani, F., Ulanovsky, A., Cairo, F., and Stroh, F.: Deep Convective Influence on the UTLS Composition in the Asian Monsoon Anticyclone Region: 2017 StratoClim Campaign Results,

10 Atmospheric Chemistry and Physics Discussions, https://doi.org/10.5194/acp-2019-1053, 2019.

Corti, T., Luo, B. P., Fu, Q., Vömel, H., and Peter, T.: The Impact of Cirrus Clouds on Tropical Troposphere-to-Stratosphere Transport, Atmospheric Chemistry and Physics, 6, 2539–2547, https://doi.org/10.5194/acp-6-2539-2006, 2006.

Höpfner, M., Ungermann, J., Borrmann, S., Wagner, R., Spang, R., Riese, M., Stiller, G., Appel, O., Batenburg, A. M., Bucci, S., Cairo, F., Dragoneas, A., Friedl-Vallon, F., Hünig, A., Johansson, S., Krasauskas, L., Legras, B., Leisner, T., Mahnke, C., Möhler, O., Molleker,

15 S., Müller, R., Neubert, T., Orphal, J., Preusse, P., Rex, M., Saathoff, H., Stroh, F., Weigel, R., and Wohltmann, I.: Ammonium Nitrate Particles Formed in Upper Troposphere from Ground Ammonia Sources during Asian Monsoons, Nature Geoscience, 12, 608–612, https://doi.org/10.1038/s41561-019-0385-8, 2019.

Joseph, B. and Legras, B.: Relation between Kinematic Boundaries, Stirring, and Barriers for the Antarctic Polar Vortex, Journal of the Atmospheric Sciences, 59, 1198–1212, https://doi.org/10.1175/1520-0469(2002)059<1198:RBKBSA>2.0.CO;2, 2002.

20 Legras, B., Pisso, I., Berthet, G., and Lefèvre, F.: Variability of the Lagrangian Turbulent Diffusion in the Lower Stratosphere, Atmospheric Chemistry and Physics, 5, 1605–1622, https://doi.org/10.5194/acp-5-1605-2005, 2005.

Pan, L. L., Honomichl, S. B., Kinnison, D. E., Abalos, M., Randel, W. J., Bergman, J. W., and Bian, J.: Transport of Chemical Tracers from the Boundary Layer to Stratosphere Associated with the Dynamics of the Asian Summer Monsoon, Journal of Geophysical Research: Atmospheres, 121, 14,159–14,174, https://doi.org/10.1002/2016JD025616, 2016.

25 Park, M., Randel, W. J., Emmons, L. K., and Livesey, N. J.: Transport Pathways of Carbon Monoxide in the Asian Summer Monsoon Diagnosed from Model of Ozone and Related Tracers (MOZART), Journal of Geophysical Research, 114, D08 303–D08 303, https://doi.org/10.1029/2008JD010621, 2009.

Pisso, I., Marécal, V., Legras, B., and Berthet, G.: Sensitivity of Ensemble Lagrangian Reconstructions to Assimilated Wind Time Step Resolution, Atmospheric Chemistry and Physics, 10, 3155–3162, https://doi.org/10.5194/acp-10-3155-2010, 2010.

30 Ploeger, F., Konopka, P., Walker, K., and Riese, M.: Quantifying Pollution Transport from the Asian Monsoon Anticyclone into the Lower Stratosphere, Atmospheric Chemistry and Physics, 17, 7055–7066, https://doi.org/10.5194/acp-17-7055-2017, 2017.

Randel, W. J., Park, M., Emmons, L., Kinnison, D., Bernath, P., a Walker, K., Boone, C., and Pumphrey, H.: Asian Monsoon Transport of Pollution to the Stratosphere., Science, 328, 611–613, https://doi.org/10.1126/science.1182274, 2010.

---

## Author Comment (AC2) · 19 May 2020

**Answer to reviewer #2**

B. Legras and S. Bucci

**Correspondence:** bernard.legras@lmd.ens.fr

We thank reviewer #2 for his/her comments.

**Answer to general comment**

**0.1 On the overall sloppiness in the layout and presentation**

As the opinion of the second reviewer is quite contrasted with that of the first reviewer who wrote that the paper is well-written,
we beg the second reviewer to forgive our poor English style but we disagree that our paper is not constructed. We worked on
the text to improve the readability of the manuscript and, in particular, to rework the single sentence paragraphs.

**0.2 On the robustness of the results**

We are aware of the possible impact of the duration of numerical calculations on the statistics built from time integrals. The
2-month integration time has been precisely chosen to avoid such effects and all the displayed results are robust in this respect.
The impact is shown in Fig. 5 and in Fig. S5, as a function of age, after normalization on each level. As these figures use a
logarithmic scale, it is clear that very little contribution to low order moment statistics can arise from ages larger than two
months except at the top levels above 380 K for EIZ and above 400 K for the other cases. In order to show this better, Fig. A2-1
compares, for the four cases of Fig. 5, the mean ages obtained by averaging over the two months period and after applying the
exponential tail correction of Scheele et al. (2005),

$$\mathcal{A}_c = \frac{\mathcal{A} + \frac{F_f}{b}\left(\tau_f + \frac{1}{b}\right)}{1 + \frac{F_b}{b}}, \tag{1}$$

where $\mathcal{A}$ is the uncorrected mean age, $\mathcal{A} = \sum_{i=0}^{P-1} F_i \tau_i \delta\tau$, calculated from the normalized impact histogram $\{F_i\}$ over $P$ bins
of width $\delta\tau$ in the age between 0 and $\tau_f$, and $-b$ is the slope of the exponential tail. Here the slope is calculated for all the
levels between $\tau = 50\,\mathrm{days}$ and $\tau = 62\,\mathrm{days}$ and its value is averaged between 330 K and 390 K.

Figure A2-1 shows that, for all cases but EIZ, the correction is fully negligeable up to 400 K and is still small at 420 K. The
results described from Sec. 3.3 onward are mostly limited to below 400 K and to the diabatic cases. EAZ results are used in

[Figure]

**Figure A2-1.** Uncorrected mean age (blue) and corrected mean age using (1) (orange) as a function of potential temperature for the four cases EAD (ERA5 diabatic), EAZ (ERA5 kinematic), EID (ERA-Interim diabatic) and EIZ (ERA-Interim kinematic).

the supplement up to 400 K and EIZ is used in Fig. S10 but to show differences between kinematic and diabatic transport in the Hadley-Walker domain, that is at low altitudes. It is also clear that estimates of the ascent rate following the modal peak are not affected, even at 420 K. Figure A2-1 has been added to the Supplement.

Figure 6 was meant to show the difficulties that arise when using average metrics. As it is unecessary to the paper and is a
5    source of confusion, it has been removed.

**0.3    On the inconsistency of vertical motion with observed clouds**

The reviewer points here to an important limitation of our approach that combines analysed velocities and heating rates with observed clouds. In the present state of the art, the reanalysis do not assimilate any cloud observations (besides winds derived from cloud motion) and therefore the model clouds are generated by the model parametrization. Although extensive compar-
10    isons demonstrate the ability of NWP models to predict the cloud cover and precipitation, observed clouds and model clouds most generally do not coincide at local scale in timing, position and extent. Therefore the cloud effects on vertical and horizontal transport also differ in the model and in the real atmosphere. Our approach assumes that the relaxation time of such effects which is up to several days in the TTL, is large enough, with respect to cloud time scale, to ensure a statistical smoothing of the cloud effects that brings the transport properties of the model close to the observations. This assumption has been used
15    already in a number of previous works (e.g. Pfister et al., 2001; Luo and Rossow, 2004; James et al., 2008; Bergman et al.,

2012; Ueyama et al., 2014; Tissier and Legras, 2016). Our present approach has been recently tested in the context of the Asian monsoon by a detailed comparison of Lagrangian trajectories with airborne tracer data (Bucci et al., 2019).

**0.4 On the usage of acronyms**

We agree that the usage of non standard acronyms should be limited but they are somewhat necessary when the same objects are mentioned in multiple instances within a text. We use here four specific acronyms to designate the reanalyses which are each used between 20 and 30 times in the main text and the supplement and the word FullAMA that denote our main region encompassing the Asian Monsoon Anticyclone is used about 60 times. We do not think that replacing them by expanded expression would improve the manuscript. We have however expanded the EAD, EAZ, EID and EIZ acronyms in most of the captions. The AMACore acronym which was not necessary has been removed.

**0.5 On the 1-D model**

We consider that the 1-D model is an important part of the paper as it fulfils the demonstration that the average transport properties of the Asian Monsoon Anticyclone can be quantitatively reduced to a small set of simple parameters. Without that part, we think that the paper will be considerably weakened.

**0.6 Minor comments**

We have made all the required changes.

**References**

Bergman, J. W., Jensen, E. J., Pfister, L., and Yang, Q.: Seasonal Differences of Vertical-Transport Efficiency in the Tropical Tropopause Layer: On the Interplay between Tropical Deep Convection, Large-Scale Vertical Ascent, and Horizontal Circulations, Journal of Geophysical Research, 117, D05 302, https://doi.org/10.1029/2011JD016992, 2012.

5 Bucci, S., Legras, B., Sellitto, P., D'Amato, F., Viciani, S., Montori, A., Chiarugi, A., Ravegnani, F., Ulanovsky, A., Cairo, F., and Stroh, F.: Deep Convective Influence on the UTLS Composition in the Asian Monsoon Anticyclone Region: 2017 StratoClim Campaign Results, Atmospheric Chemistry and Physics Discussions, https://doi.org/10.5194/acp-2019-1053, 2019.

James, R., Bonazzola, M., Legras, B., Surbled, K., and Fueglistaler, S.: Water Vapor Transport and Dehydration above Convective Outflow during Asian Monsoon, Geophysical Research Letters, 35, L20 810, https://doi.org/10.1029/2008GL035441, 2008.

10 Luo, Z. and Rossow, W. B.: Characterizing Tropical Cirrus Life Cycle, Evolution, and Interaction with Upper-Tropospheric Water Vapor Using Lagrangian Trajectory Analysis of Satellite Observations, Journal of climate, 17, 4541–4563, 2004.

Pfister, L., Selkirk, H. B., Jensen, E. J., Schoeberl, M. R., Toon, O. B., Browell, E. V., Grant, W. B., Gary, B., Mahoney, M. J., Bui, T. V., and Hintsa, E.: Aircraft Observations of Thin Cirrus Clouds near the Tropical Tropopause, Journal of Geophysical Research: Atmospheres, 106, 9765–9786, https://doi.org/10.1029/2000JD900648, 2001.

15 Scheele, M. P., Siegmund, P. C., and Velthoven, P. F. J.: Stratospheric Age of Air Computed with Trajectories Based on Various 3D-Var and 4D-Var Data Sets, Atmospheric Chemistry and Physics, 5, 1–7, https://doi.org/10.5194/acp-5-1-2005, 2005.

Tissier, A.-S. and Legras, B.: Convective Sources of Trajectories Traversing the Tropical Tropopause Layer, Atmospheric Chemistry and Physics, 16, 3383–3398, https://doi.org/10.5194/acp-16-3383-2016, 2016.

Ueyama, R., Jensen, E. J., Pfister, L., Diskin, G. S., Bui, T. P., and Dean-Day, J. M.: Dehydration in the Tropical Tropopause Layer: A
20 Case Study for Model Evaluation Using Aircraft Observations: A Case Study of TTL Dehydration, Journal of Geophysical Research: Atmospheres, 119, 5299–5316, https://doi.org/10.1002/2013JD021381, 2014.